# Coseismic Surface Rupture Probabilities from Earthquake Cycle Simulations: Influence of Fault Geometry

Octavi Gómez-Novell[1], Francesco Visini[2], José Antonio Álvarez-Gómez[3], Bruno Pace[4], Julián García-Mayordomo[1]

[1] Instituto Geológico y Minero de España, IGME, CSIC, Madrid, Spain
[2] Istituto Nazionale di Geofisica e Vulcanologia, Chieti, Italy
[3] Universidad Complutense de Madrid, Madrid, Spain
[4] Università Degli Studi "Gabriele d'Annunzio" di Chieti e Pescara, Chieti, Italy

*Correspondence to*: Octavi Gómez-Novell (ogomez.novell@igme.es)

**Abstract.** Earthquake surface ruptures are a significant hazard for critical infrastructure and society. Probabilistic Fault Displacement Hazard Analysis (PFDHA) uses empirical and numerical models to estimate the surface rupture likelihood as the first component. However, empirical datasets are often incomplete and limited to few geodynamic settings, reducing their accuracy for site-specific analyses. Moreover, existing models do not capture the influence of physical fault parameters, such as geometry, on surface rupture occurrence nor its spatial variability. We use the RSQSim rate-and-state earthquake simulator to simulate seismicity across twelve alternative geometries of a test fault that incorporate variations of fault connectivity at depth, dip and fault trace sinuosity, aiming for a systematic evaluation of their influence on the probability of primary surface rupture and its spatial variability. Our results show that fault geometry is key in controlling the probability of surface rupture. Models with fault connectivity at depth and greater fault trace sinuosity yield higher probabilities than their counterparts. Conversely, disconnected models limit rupture propagation across segments, reducing surface rupture capability in specific fault regions. This study demonstrates the importance of considering fault geometry when assessing seismic hazards and confirms that earthquake cycle simulators offer a robust tool for next generation PFDHA models.

## 1 Introduction

Earthquake surface ruptures are a phenomenon that represents a critical hazard for infrastructures such as pipelines, bridges and dams, and play a key role in shaping tectonically active landscapes. As such, forecasting the likelihood of and the expected surface displacement are essential components of hazard assessment strategies, especially for critical infrastructures. This is evidenced by the fact that surface fault displacement hazard is currently well integrated in several international safety frameworks (FEMA, 2015; IAEA, 2019, 2021, 2022, 2025; Valentini et al., 2025a, b)

Probabilistic Fault Displacement Hazard Analysis (PFDHA), introduced by Youngs et al. (2003), is a methodology designed to estimate the likelihood of surface rupture and displacement expected in a site or region. Over time, most PFDHA approaches have relied on developing new and more updated empirical models that incorporate earthquake datasets in different tectonic

environments as they become available (e.g., Moss et al., 2013, 2024; Pizza et al., 2023; Takao et al., 2013; Visini et al., 2025; Yang et al., 2021).

Empirical data shows that surface rupture is generally correlated with earthquake magnitude and, as such, the probability of damage related to fault displacement at surface increases accordingly with magnitude. However, empirical models still present key limitations. First, the earthquake datasets used to derive regressions are geographically sparse and heterogeneously distributed across regions and tectonic environments. As such, models usually incorporate data from multiple regions, making them less accurate for site-specific purposes. These models, in fact, highlight the importance of incorporating site or region-specific elements like faulting style or soil conditions for the probability of surface rupture. Second, current empirical models do not consider spatial variability of fault displacement further limiting accuracy in site-specific applications. Recent advances have introduced numerical methods into PFDHA (e.g., Mammarella et al., 2024). These numerical solutions tackle intrinsic limitations on site-specific applicability of empirical approaches and do not require empirical earthquake datasets to develop the regressions.

Despite the empirical models have been and are currently widely used, along with the numerical solutions, they lack the capacity to incorporate and fully capture the physical parameters that govern fault ruptures and displacement. In this context, physics-based earthquake cycle simulators, being model-driven approaches, could improve forecasting capabilities for PFDHA (e.g., Valentini et al., 2025b). These simulators are algorithms that incorporate frictional and stress evolution laws governing the seismic cycle to simulate seismicity in pre-defined fault systems. For one, this allows simulating earthquake ruptures and rupture patterns consistently with observations (e.g., Richards-Dinger and Dieterich, 2012; Zielke and Mai, 2023). For another, the simulators generate large earthquake catalogues over many earthquake cycles, allowing robust statistical data exploitation capabilities for probabilistic-based approaches like PFDHA. Recently, Daglish et al. (2025) developed a novel PFDHA application that combines, for the first time, historical earthquake data and physics-based earthquake cycle simulations to evaluate road exposure and vulnerability to fault displacement in New Zealand. In detail, the authors use the New Zealand community fault model to simulate earthquake ruptures and, thus, to generate fault surface displacement fields from the on-fault simulated displacements, which are then used to evaluate their impact in the road network of the country. However, so far, no study has used earthquake cycle simulators to systematically evaluate the impact of fault-specific parameters, such as multiple realizations of fault geometries, into the probability of surface rupture, including their spatial variability. We hypothesize that these simulators can enhance PFDHA by allowing to model the impact of fault-specific characteristics like fault geometry, a more accurate representation of fault ruptures, and the inclusion of many seismic cycles to improve statistical representativeness.

In this study, we explore the feasibility of integrating earthquake cycle simulations into PFDHA using RSQSim, a rate-and-state earthquake cycle simulator (Richards-Dinger and Dieterich, 2012). We examine how variations in fault geometry—including segmentation, dip variability, and along-strike trace sinuosity (e.g., fault roughness linked to the traces at the top and

bottom of the fault)—affect surface rupture probability on the principal fault. Specifically, we undertake this work in a case study at the Mt. Vettore Fault in Central Italy (Fig. 1).

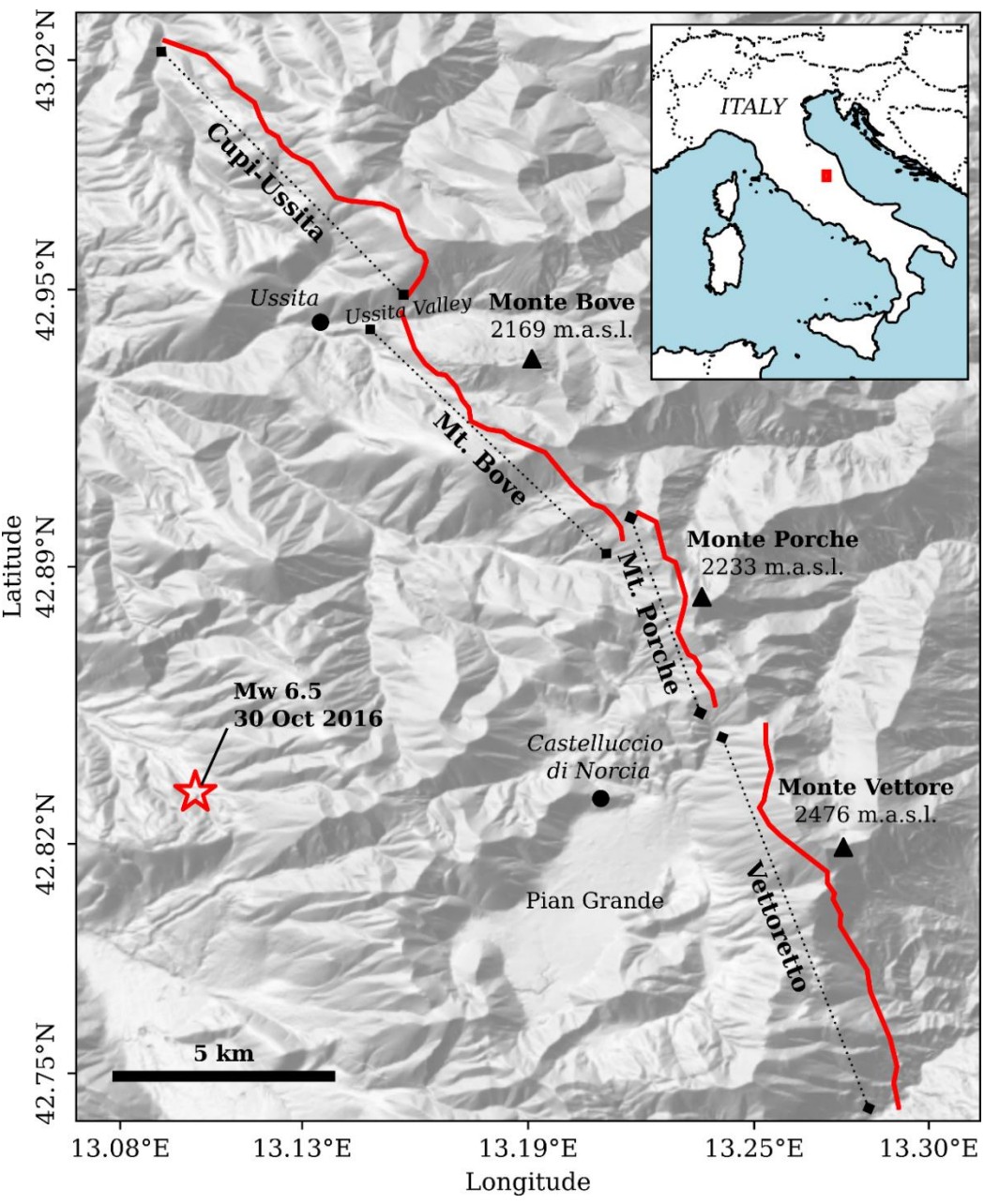

Figure 1. Fault trace of the Monte Vettore Fault System (MVFS). Fault traces correspond to the "Fault" level as compiled in the Central Apennines Database (CAD); Faure-Walker et al. (2021). The fault level considers first-order structures – i.e., fault segments – that have the potential of rupturing entirely but that have prominent end boundaries

**that are considered as potential rupture barriers. The base hillshade has been created from the 10-m resolution TINITALY 1.1 digital elevation model (Tarquini et al., 2023).**

The Mt. Vettore Fault is a normal fault in the Central Apennines that was the source of the 2016 $M_w$ 6.5 Central Italy earthquake. The activity of the fault is geologically well known, with several studies focusing on the characterization of its surface morphology, paleoseismic activity and seismotectonics (e.g., Cinti et al., 2019; Galli et al., 2019; Lavecchia et al., 2016; Puliti et al., 2020). Despite this, there is little consensus on a preferred subsurface geometry model for the fault, which makes it a suitable candidate to explore geometric implications for earthquake cycle models and to confront our results with

geological observations in the area.

Our objectives are: 1) to assess the impact of fault geometry features such as depth connectivity, dip and trace sinuosity on primary surface rupture occurrence; 2) to evaluate the consistency between simulation-based probabilities and existing empirical and numerical models, as well as with geological observations; and 3) to characterize the spatial variability of rupture probability along the fault trace. Our outcomes aim to advance the use of physics-based models in fault displacement hazard

assessments, contributing to a more robust and fault-specific PFDHA methodological framework.

## 2 Methods and modeling setup

In this section we explain the basic formulation and assumptions of the RSQSim simulator (Richards-Dinger and Dieterich, 2012) and the modeling setup followed in the earthquake cycle simulations, including the fault geometric models explored and the initial parameter selection. We also detail the statistical approaches we conducted during the analyses of the simulation

outputs to derive surface rupture statistics for each one of the model setups, as well as to compare with observations.

### 2.1. The RSQSim simulator

RSQSim is an earthquake cycle simulator that employs the rate-and-state friction (RSF) laws first introduced by Dieterich (1979) and later works by Ruina (1983), Tullis (1988) or Marone (1998) to model long-term earthquake catalogues in predefined boundary-element fault geometries. In the RSF constitutive law,

$$\tau^{frict} = \sigma \left[ \mu_0 + a ln \left( \frac{V}{V_0} \right) + b ln \left( \frac{\theta V}{D_c} \right) \right] \tag{1}$$

$T^{frict}$ is the shear stress-resisting motion, $\sigma$ is the normal stress, $\mu_0$ is the steady state friction coefficient, $V$ is the slip speed, $V_0$ is the reference slip speed, $\theta$ is the state variable, $D_c$ is the characteristic slip distance, and $a$ and $b$ are the direct- and evolution effect parameters of the RSF law, respectively.

In RSF, the relationship between $a$ and $b$ coefficients *(a-b)* determines the fault slip behavior. *(a-b)* < 0 implies velocity

weakening behavior, which is observed in most seismic slip – i.e., stick-slip. Conversely, *(a-b)*>0 implies velocity strengthening, which is related to stable sliding – i.e., fault creep.

RSQSim works with discretized fault surfaces in the sense of Rice (1993) and the stresses are analyzed for each fault element throughout the whole computation. To model the seismic cycle, RSQSim considers three states – healing, nucleation and seismic rupture – that employ analytical equations to resolve the evolution of stresses, slip speed and the state variable at every state (Richards-Dinger and Dieterich, 2012).

Although RSQSim is not a fully dynamic rupture simulator, the code incorporates approximations to elasto-dynamics that allow to model the coseismic phase of the earthquake cycle realistically. These approximations include reduction of the direct effect constant of the RSF law $a$ in slipping patches and the inclusion of a dynamic overshoot coefficient, which replicates the dynamic overshoot from fully dynamic rupture simulations (Richards-Dinger and Dieterich, 2012).

All these features make RSQSim a strong tool to model not only simulated seismic sequences over many seismic cycles, but also to statistically analyze fault rupture patterns and fault interaction in fault systems at the earthquake cycle scale.

## 2.2. Fault geometry and trace sinuosity

We defined a set of fault plane geometric models that increase complexity in both fault connectivity and dip variability at depth, and along-strike trace sinuosity (Fig. 2). Fault trace sinuosity refers to the curvature of the fault traces at both the surface and base of the seismogenic thickness, which result in fault plane geometric roughness. We use a detailed fault mesh with 300m-side triangular fault elements to better capture geometric fault trace complexities. In all models the seismogenic depth is set at 12 km, which is consistent with studies in the region (e.g., Lavecchia et al., 2016).

In terms of fault connectivity, we define three levels. The simplest one is based on the segmentation of the Mt. Vettore fault, as proposed in the Fault2SHA Central Apennines Database (Faure Walker et al., 2021; Fig. 1). The fault model consists of four fault surfaces completely disconnected from the top to the bottom of the seismogenic layer and with a constant dip of 60° (Fig. 2). This model assumes that the observed surface segmentation is preserved at depth. The second level assumes that the fault surfaces observed from the segmentation link at depth at 7 km, which is constistent with interpretations of the Mt. Vettore by Lavecchia et al. (2016). Like in the previous model, the dip is constant throughout the whole seismogenic layer at 60°. The third level has the same characteristics as the second one in terms of linkage at depth, but with a listric geometry, from 60° to 30° between 7 and 12 km. The listric geometry at these depths is also consistent with modeling data by Lavecchia et al. (2016). The three levels are referred to as D for Disconnected, C for Connected constant and L Connected listric, respectively, throughout the text.

In terms of along-strike trace sinuosity, we consider four levels to capture the complexity in the fault surface. The first level is defined by fully linear fault traces at top and bottom, which define fully planar fault surfaces. From the second to fourth levels the surface trace is based on the mapped fault trace at surface from the CAD Database (Fig. 1), with smoothing to remove details below resolution of the 300m fault elements, while the bottom trace changes with different levels of complexity. The second level considers a straight trace at depth, the third one considers a smoothed fault trace at depth, and the fourth preserves

the surface fault trace at depth. The four sinousity levels are defined as increasing numeric values for increasing sinuosity of both top and bottom fault trace. From no sinuosity to maximum sinuosity these four levels are 0, 0.3, 0.6 and 1, respectively.

By combining all these levels, we define a matrix of 12 fault models that we explore in our study (Fig. 2).

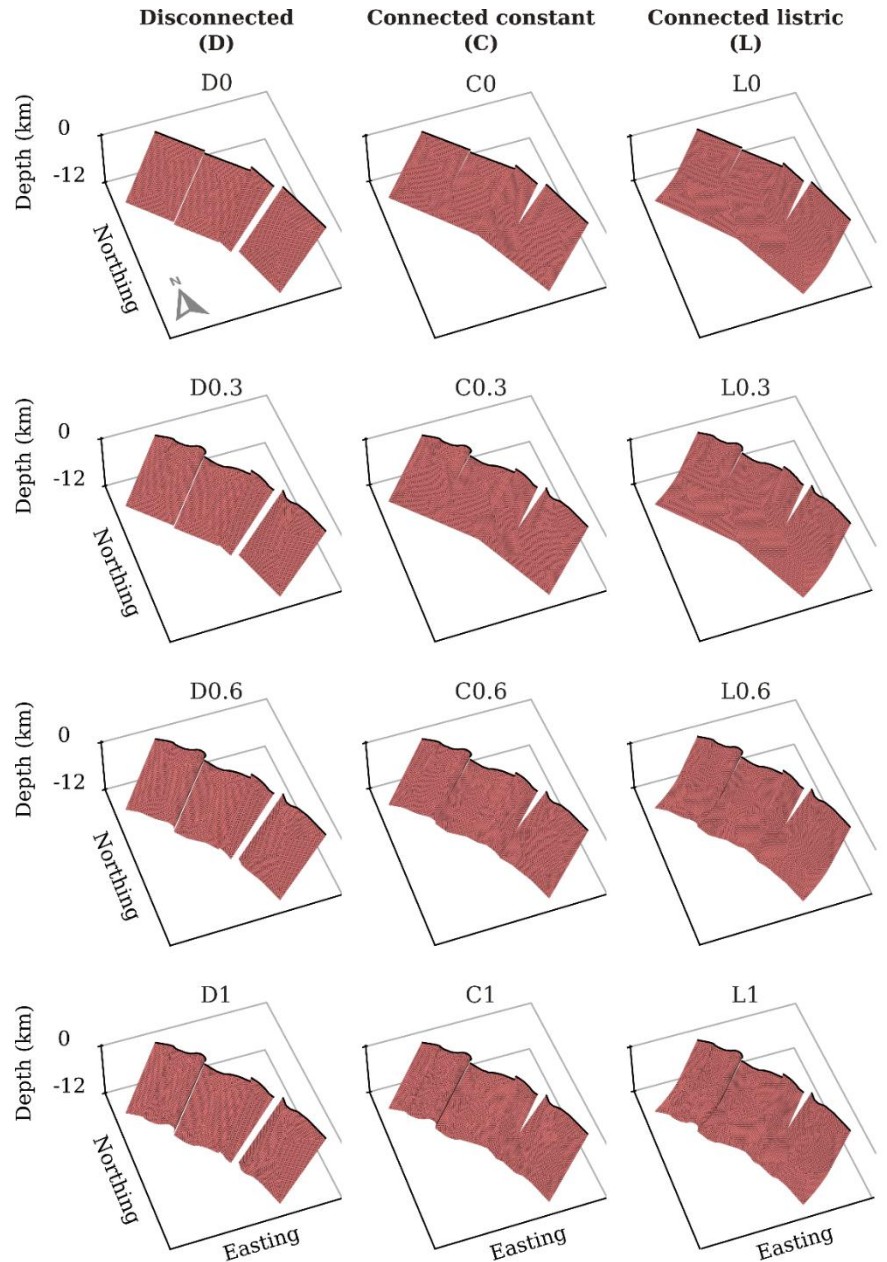

**Figure 2. Geometric fault models explored in this study. Columns group models with the same fault connectivity and dip at depth, while rows group models with the same fault sinuosity defined by the fault surface and bottom trace.**

## 2.3. Model parameters

### 2.3.1. Initial stresses

The initial stresses that RSQSim uses for the modeling are the effective normal and shear stresses $\sigma'_n$ and $\tau$, respectively, related through a static friction law: $\tau = \mu\sigma'_n$, where the $\mu$ is the friction coefficient. The effective normal stress is computed through the relationship between the principal stresses $(\sigma'_1)$ and $(\sigma'_3)$. In an extensional regime, the maximum principal stress $\sigma'_1$ follows a gravitational profile $\sigma'_1 = \rho g z$, where $\rho$ is the rock density (2600 kg/m$^3$ in this study considering a lithostatic gradient of 26 MPa/km in the Amatrice region; Montone and Mariucci, 2020), $g$ is the gravitational acceleration and $z$ is depth.

As such, the minimum principal stress $\sigma'_3$ can be derived from the formulations by Sibson (1985)

$$\frac{\sigma'_1}{\sigma'_3} = \frac{(1+\mu\cot\theta)}{(1-\mu\tan\theta)} \qquad (2)$$

where $\theta$ is 90° minus the dip angle in an extensional setting. Following Liao et al. (2024), from these equations we can compute the initial and shear stresses on the fault plane (equations 3 and 4, respectively).

$$\sigma'_n = \frac{\sigma'_1+\sigma'_3}{2} + \frac{\sigma'_1-\sigma'_3}{2}\cos 2\theta \qquad (3)$$

$$\tau = \frac{\sigma'_1-\sigma'_3}{2}\sin 2\theta \qquad (4)$$

For this study, the initial stresses are uniform throughout the fault surfaces, computed for a depth equal to the half of the seismogenic thickness (6 km). The initial $\sigma_n'$ is set at 129.5 MPa and the initial $\tau$ at 45.9 MPa. The friction coefficient considered is 0.6.

### 2.3.2. Frictional parameters

Frictional parameters have a considerable impact on the simulated catalogues with RSQSim, especially the (*a-b*) relationship of the rate and state law. To objectively select the most suitable parameters, we employ a benchmarking method recently developed by Gómez-Novell et al. (2025a). This approach ranks the performance of simulated earthquake catalogues by quantifying their fit to a series of empirical benchmarks, namely earthquake scaling relationships and the shape of a target magnitude-frequency distribution with a b-value of 1.

To find the most suitable set of (*a-b*) parameters, we run benchmarking tests for the two geometric end members (D0 and L1) using an exploration parameter tree, resulting in 11 models for each geometric configuration (Table 1). From these benchmark tests we obtain quality ranks for both geometric ends as a function of (*a-b*). To find the overall best model, we average both model ranks for each (*a-b*) combination and select the better performing one. In table 1 we show the resulting ranks of each (*a-b*) combination in a 0-1 scale (0 being the best and 1 the worst). The selected a and b values are 0.001 and 0.004, respectively.

| a | b | (a-b) | D0 | L1 | Final Rank (mean) |
|---|---|---|---|---|---|
| 0.001 | 0.002 | -0.001 | 0.11 | 0.25 | 0.18 |
| 0.001 | 0.003 | -0.002 | 0 | 0.05 | 0.03 |
| 0.001 | 0.004 | -0.003 | 0.007 | 0 | 0 |
| 0.001 | 0.005 | -0.004 | 0.18 | 0.16 | 0.17 |
| 0.001 | 0.006 | -0.005 | 0.36 | 0.41 | 0.39 |
| 0.001 | 0.007 | -0.006 | 0.23 | 0.21 | 0.22 |
| 0.001 | 0.008 | -0.007 | 0.51 | 0.44 | 0.48 |
| 0.001 | 0.009 | -0.008 | 0.51 | 0.38 | 0.45 |
| 0.001 | 0.01 | -0.009 | 0.24 | 0.51 | 0.38 |
| 0.01 | 0.015 | -0.005 | 1 | 1 | 1 |
| 0.001 | 0.015 | -0.014 | 0.95 | 0.85 | 0.9 |

**Table 1. Benchmarking ranks for the two end-member fault geometric models considering different rate-and-state coefficients (*a* and *b*), and the mean final rank for both.**

For the remaining input parameters such as $V$, $V_0$, $\theta$ and $D_c$ of the RSF law we use the default values defined by RSQSim. Check the full parameter list of the simulated catalogues in the data repository of this publication (Gómez-Novell et al., 2025b).

### 2.3.3. Fault slip rates

RSQSim requires to prescribe slip rates to each one of the fault elements for the simulation. In this study we use a tapered slip rate distribution throughout the fault plane (Fig. 3) to minimize stress singularities at the fault edges generated by the back-slip loading approach that RSQSim employs (see Shaw, 2019). Our customized slip rate distributions are tapered in an elongated concentric shape across the whole fault, considering that the maximum slip rate is reached towards the bottom half of the seismogenic thickness (between 8-9 km), following the example from Delogkos et al. (2023). Unlike Delogkos et al. (2023), our slip rate distribution is tapered across the whole fault instead of the fault segment for several reasons. First, to ensure consistency across models and avoid large differences in slip rate distributions between disconnected and connected cases, which could introduce variability between catalogues and obscure strictly geometry-related effects. Second, to better test the control of fault geometry. That is, to better test whether the observed variability patterns in slip along-strike (e.g., general slip reduction at the fault segment tips; see section 4.4) can emerge in our models from fault geometry alone, rather than from a pre-imposed slip rate profile. Third, to guarantee consistency with higher order structure because, despite the Mt. Vettore fault is segmented, it is widely regarded as a single fault structure in literature (see Faure Walker et al., 2021).

We ensure that surface slip rate in the central part of the trace is consistent with the geological slip rates derived from surface studies in the central part of the fault – around 0.9 mm·yr$^{-1}$, which is the upper bound of the minimum slip rate (Pousse-Beltran et al., 2022) (Fig. 3b).

The average slip rate of all fault elements is 1 mm·yr$^{-1}$. This value is the median between the lower and upper bounds of the minimum and maximum slip rates estimated by Pousse-Beltran et al. (2022), respectively: 0.6 - 1.4 mm·yr$^{-1}$ (0.7 + 0.2/−0.1 mm·yr$^{-1}$ and 1.2±0.2 mm·yr$^{-1}$). Supplement figures S1 and S2 show the input slip rate data considered for each model.

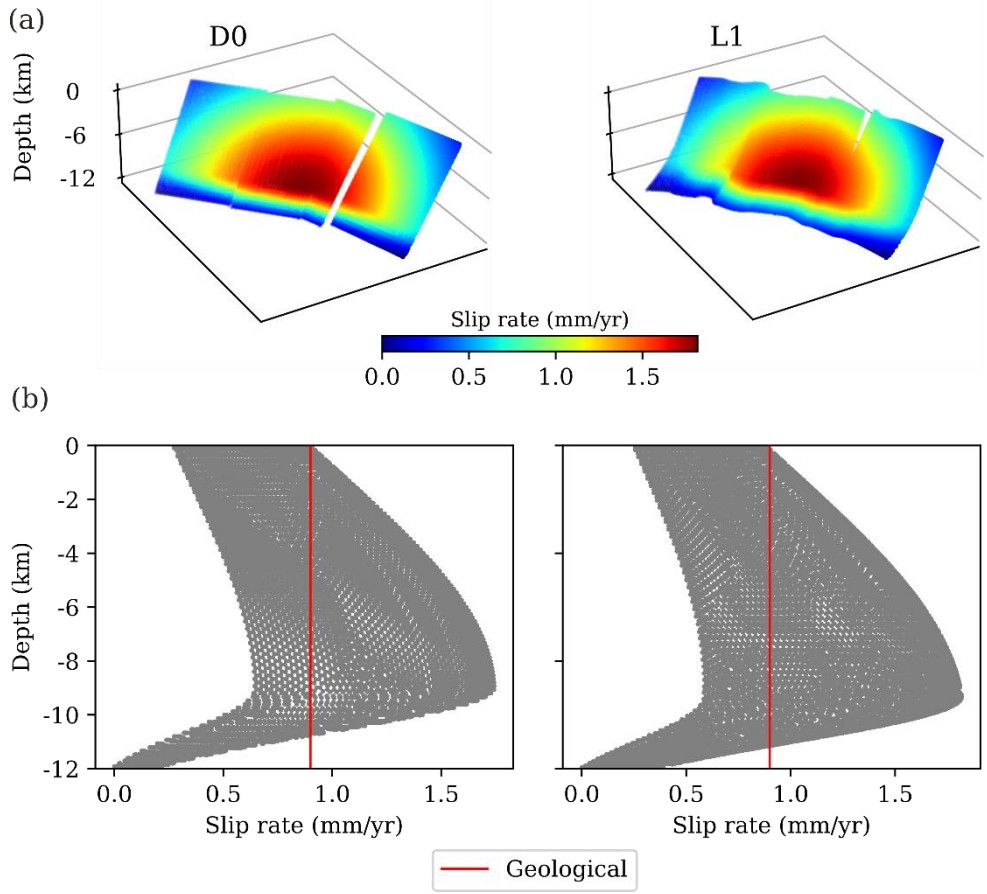

**Figure 3. a) Slip rate distribution on the fault plane for the two end-member models of fault connectivity and sinuosity explored in this study. b) Down-dip slip rate distribution for the same end-member models as in panel (a). Each grey point of the graph corresponds to the slip rate value of a fault element in the model. The vertical red line indicates the upper slip rate bound of the minimum surface slip rate estimated for the Mt. Vettore fault in its central segment (0.9 mm·yr$^{-1}$) by Pousse-Beltran et al. (2022). The average slip rate considering all fault elements is 1 mm·yr$^{-1}$. Slip rate distributions of all twelve models considered in the study are shown in the Supplement figures S1 and S2.**

## 2.4. Simulations

RSQSim models are run for 100,000 simulated years, of which we analyze only the last 50,000 years to ensure that the initial stresses have evolved sufficiently from the initial conditions and that the earthquake cycle has stabilized. We also perform a magnitude completeness analysis to remove earthquakes below that magnitude following the maximum curvature approach, and we also remove events that involve less than 10 fault elements. This is done to ensure that the rupture process of the simulated events is well resolved and thus the analyzed magnitudes are those that fit the best the empirical relations (e.g., see Zielke and Mai, 2023).

## 2.5. Catalogue analysis

The probabilities of surface rupture are obtained by filtering the catalogue data on the fault elements located at the surface analyzed with respect to the data registered throughout the whole fault. The likelihood of surface rupture is computed for the whole fault as the quotient between the number of events of magnitude $m$ equal to a reference magnitude $M$ ($m=M$) reaching the surface and the total number of events of $m=M$ occurred in the fault (light grey and dark grey histograms, respectively in Figure 4a). Surface ruptures are accounted regardless of the number of surface fault elements involved or their slip. Then, we compute the conditional probability of surface rupture for m=M at the whole fault by fitting a logistic regression to the discrete data points, as introduced by Youngs et al. (2003)

$$P = \frac{e^{a+b*M}}{1+e^{a+b*M}} \qquad (5)$$

where P is the conditional probability of primary surface rupture, and $a$ and $b$ are the fit coefficients of the regression, not to be confused with the RSF law coefficients.

We also analyze the probability of surface rupture variations along strike. For each fault element at surface, we compute the ratio between the number of events of m ≥M affecting that patch and the total number of events of $m \geq M$ in the whole fault. In figure 4b the dotted line is the number of events of $m \geq M$ in each patch at the surface along strike, while the red line represents the total number of events for $m \geq M$ in the whole fault (reported as a line for better visualization).

For the different surface rupture probability analyses, we use different magnitude thresholds. Regressions are computed for Mw ≥4.0 to ensure all magnitudes are considered without biasing the regressions, including those with lower probability. Along-strike surface rupture probabilities are computed for Mw ≥ 6.0 as these magnitudes have larger probability of surface rupture and, thus, ensure a better visualization of the spatial variability patterns along strike and the effect of fault segmentation.

We compare the regressions of surface rupture probability with the empirical curves for normal faults in the Great Basin by Youngs et al. (2003) and worldwide normal faults by Pizza et al. (2023), and the numerical curves for normal faults by Mammarella et al. (2024), considering the Thingbaijam et al. (2017) scaling equations and site-specific inputs for our study

area: a seismogenic depth of 12±1 km, a dip angle of 60±5° and the hypocentral depth ratio of Italian normal faults (i.e., peak
of hypocentral depths at around 67% of the seismogenic depth). See Mammarella et al. (2024) for details on input parameters.

The analysis of the catalogues is performed for their entire 50,000 year length as well as for shorter time windows to capture
potential variability in the earthquake rates (see section 3.2). All probabilities computed in the paper refer to primary surface
rupture on the principal fault; modeling secondary faulting or off-fault deformation is beyond of our scope.

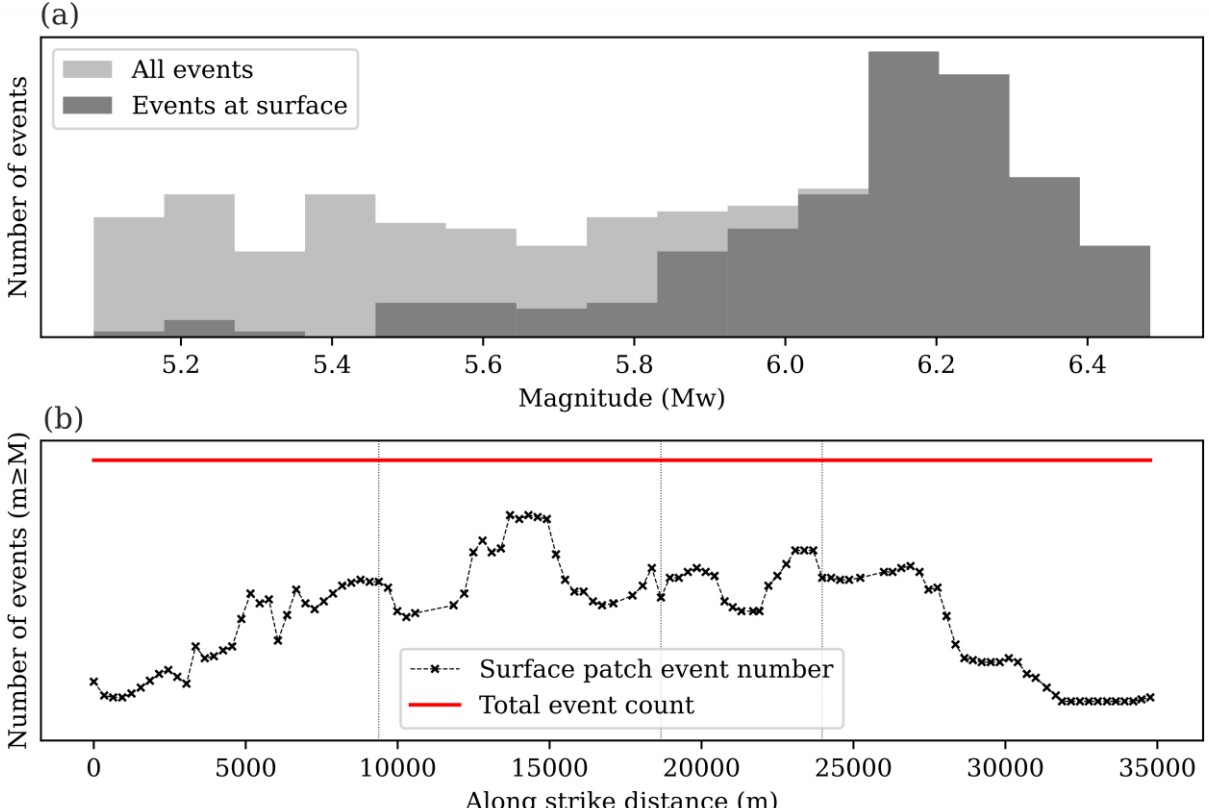

**Figure 4. Method to calculate the likelihood of surface rupture for discrete magnitudes in the a) whole fault and b)
along-strike in a given simulated catalogue. In panel (a) the likelihood of surface rupture is computed as the ratio
between the number of earthquakes of magnitude M rupturing the surface at any fault location and the total number
of events of that magnitude M occurring anywhere in the fault (rupturing the surface or not). In panel (b), the
probability of surface rupture is computed for each fault element at surface and along strike using the same ratio as
for the whole fault but considering all magnitudes m equal or larger than a given magnitude threshold M (m≥M).**

## 2.6. Comparison with observations

We quantitatively compare the outputs of the simulations with surface geological observations of coseismic slip distribution of the largest main shock of the 2016 earthquake sequence (30 october $M_w$ 6.5; Chiaraluce et al., 2017) and the cumulative

throw for the past 18 kyr (approximately the age of the Mt. Vettore fault scarp in the Central Apennines; Puliti et al., 2020). To do this, we assign each field measurement to the nearest surface fault element in the model. If multiple measurements fall within the same element, their values are averaged. Because each fault model has a slightly different geometry and mesh configuration, the assignment of field data points to fault elements can vary, leading to small differences in the coseismic and cumulative throw profiles.

The coseismic slip distribution of the 2016 $M_w$ 6.5 earthquake, obtained from the SURE 2.0 database (Nurminen et al., 2022), is compared against the net surface slip distributions along strike for all simulated earthquakes with magnitudes compatible with the 2016 coseismic magnitude. Given that the maximum magnitudes ($M_{max}$) of the simulated catalogues do not exceed $M_w$ 6.6 (see details in section 3.1.3), we use the 2016 magnitude to select such magnitudes following the condition: $M_{max}$-0.1 $\leq M \leq M_{2016} + 0.1$.

The cumulative throw along strike is obtained from the measurements reported by Puliti et al. (2020) on the main fault and compared against the cumulative throw of all events of $M_w \geq 5.5$ that rupture the surface in the simulations. Because RSQSim computes the net slip of earthquakes in the dip direction on all fault elements, we convert them to fault throw considering the dip of the elements in the fault model (60º).

In both cases, the agreement between simulated values and observations is evaluated using the mean absolute error (MAE) –

i.e., the average of the differences between model and observation.

## 3 Results

In this section we describe the main results from the simulated catalogues as well as the results from the probability of surface rupture analyses.

### 3.1. Observations on the simulated earthquake catalogues

### 3.1.1. Agreement with scaling relations

As a first step and to ensure optimal performance of the simulated earthquake catalogues for the different fault models explored, we analyze how well the simulated catalogues match the predicted values of the rupture area-seismic moment scaling relationship by Leonard (2010) for normal faults. Figure 5 shows the analysis for the two geometric end members analyzed for the model parametrization: D0 and L1. The analysis for the complete set of geometric models can be found in figure S3.

Figure 5 shows that all models have a good agreement with the scaling relationship, as all the simulated events fall within the two standard deviation range of the relationship. The agreement between model and empirical data improves in higher

magnitudes (higher rupture areas) with most events falling around the mean value of the scaling relationship. This behavior is linked to the better numerical resolution of the fault rupture process for higher rupture areas that involve a larger number of fault elements (Zielke and Mai, 2023).

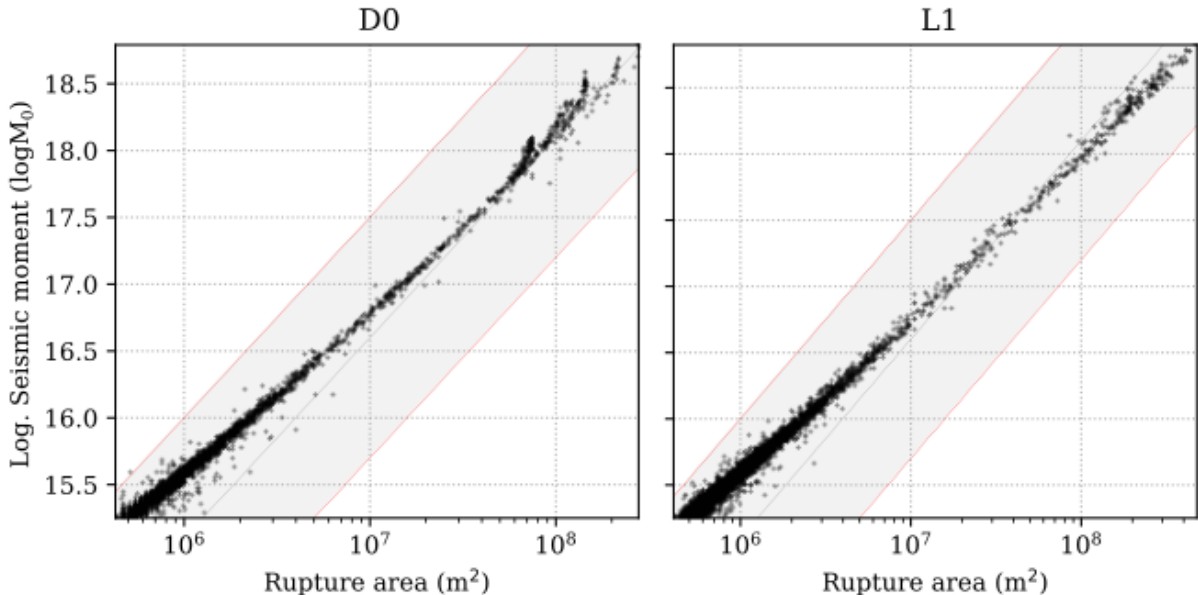


**Figure 5. Rupture area – seismic moment scaling relationship from the simulated catalogues of the two end-member models of fault connectivity and sinuosity explored in this study. These scaling relationships are plotted together with the two sigma uncertainty ranges of the rupture area – moment magnitude empirical relationship (normal faults) by Leonard (2010). Scaling relationship figures for all twelve models are shown in figure S3 of the Supplement.**


### 3.1.2. Hypocenter depth of the earthquakes

We analyze the hypocenter depth of the earthquakes in the simulations for earthquakes of $M_w{\geq}5$. The hypocenter depth distribution is primarily influenced by the geometric assumptions, particularly fault connectivity at depth. Significant changes in earthquake nucleation depth happen between the disconnected and connected models, regardless of the fault dip variability
(constant or listric). In the disconnected models, hypocenters show a depth distribution closely correlated with the slip rate depth distribution, peaking at around 9 km (Fig. 3b and Fig. 6). Conversely, in the connected models, hypocenter depth is less influenced by slip rate and more by the connection depth of the fault segments, with a peak located between 7-8 km, corresponding to the linking depth of the segments.

Regarding sinuosity, its influence on the depth hypocenter distribution is weaker than fault connectivity, but still noticeable. Introducing fault sinuosity increases the number of events $M_w \geq 5$ shallower than the peak of the hypocenter depth distribution, an effect that is more evident in connected models than the disconnected ones. For models with surface fault sinuosity, increases in trace sinuosity at the seismogenic depth lead to a more balanced, smoother, hypocenter depth distribution. Interestingly, the listric fault dip slightly reduces shallower nucleations of smaller magnitudes between $M_w < 6.0$ (Fig. 6).

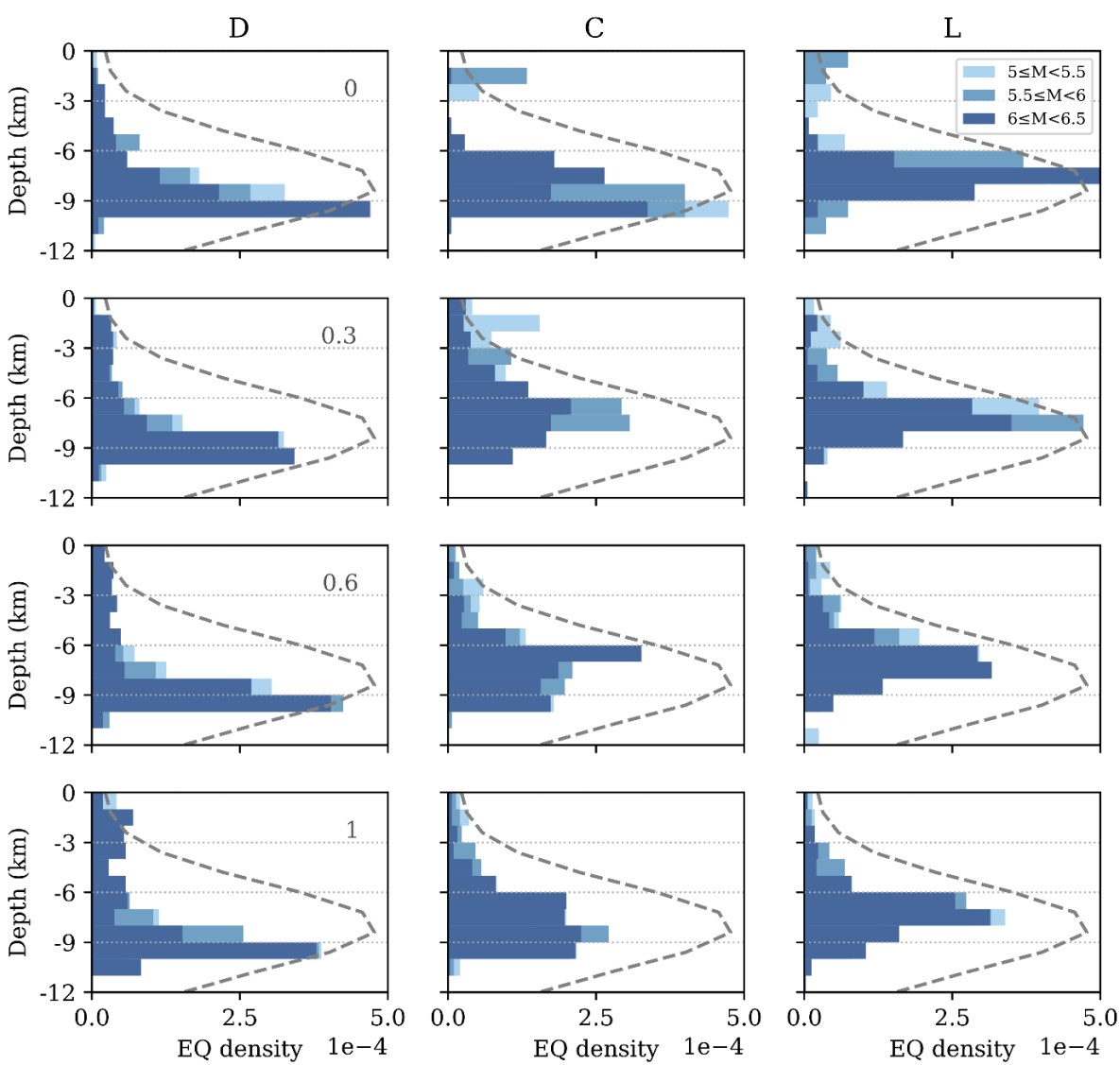

**Figure 6. Magnitude-dependent hypocenter depth distributions of the different geometric models explored in this study. The dashed curve corresponds to the empirical hypocentral depth distribution for extensional faults in Italy from Mammarella et al. (2024).**

We compare the hypocenter depth distribution of the simulations with the probability density function (PDF) of the empirical hypocenter depth distribution for normal faults in Italy from Mammarella et al. (2024) (Fig. 6). The hypocenter depth distributions modeled are generally consistent with the empirical ones for all models. Among those, the connected models, especially those with listric geometry (L), show the best agreement, with coincident peak hypocenter depths between 6 and 9 km.

The same observations apply for the hypocenter depths of smaller magnitude earthquakes of $M_w$ 4-5 (see figure S4), implying that the control of geometry in earthquake nucleation depth affects all magnitude ranges.

### 3.1.3. Magnitude-frequency distributions and maximum magnitudes

The magnitude frequency distributions (MFDs) of the simulated catalogues show a clear impact of the fault geometry into the shape of the MFDs and earthquake frequency. The shape of the MFDs is primarily controlled by the fault sinuosity. In the depth-connected models, the earthquake frequency decreases with decreased sinuosity, up to almost one order of magnitude between end-member sinuosity models (Fig. 7a). For instance, in the C model, the number of $M_w \geq 4$ events drops from around 12,000 events to 3,000 between the sinuosity level 1 and 0 models, respectively. Moreover, decreasing sinuosity deviates the MFD shape from a Gutenberg-Richter to a characteristic shape, with a pronounced event deficit in the middle magnitude range (from $M_w$ 4.5 to 6). Conversely, in the disconnected models, the impact of sinuosity is more limited. The shape of the MFD remains invariable with a slight characteristic geometry, while earthquake frequency decreases modestly with decreasing sinuosity. For instance, for $M_w \geq 4$ the earthquake number changes from 10,000 to 8,400 in the level 1 and 0 models, respectively.

The maximum magnitude is directly proportional to the fault connectivity at depth and inversely proportional to fault sinuosity. The connected models (both C and L) yield higher $M_{max}$ values overall compared to the disconnected ones, with values varying from $M_w$ 6.4 to around 6.6 in the connected models, and $M_w$ 6.3 to 6.46 in the disconnected (Fig. 7b). Among all models, the L ones are those with higher magnitudes all $M_w \geq 6.5$, which is expected given that the available area for rupture is larger. Contrarily, fault sinuosity results in a decrease of 0.1 to 0.2 magnitude units between end-member models (sinuosity levels 0 and 1) of a same depth-connectivity model (Fig. 7b). Such sinuosity introduces roughness to the fault surfaces that might attenuate stress transfer and rupture propagation, decreasing the expected maximum magnitudes.

For comparison, we evaluated whether the maximum magnitudes of the simulated catalogues reach the $M_w$ 6.5 earthquake produced by the Mt Vettore fault. None of the disconnected models reach the observed magnitude; the sinuosity 0 model yields the highest $M_{max}$ among them, at $M_w$ 6.46. In the C models, only the two with lower sinuosity, -C0 and C0.3- produce events comparable to the 2016 earthquake, with $M_{max}$ values of $M_w$ 6.59 and 6.51, respectively. In contrast, all sinuosity models in the L category, reach or exceed the magnitude of the 2016 event.

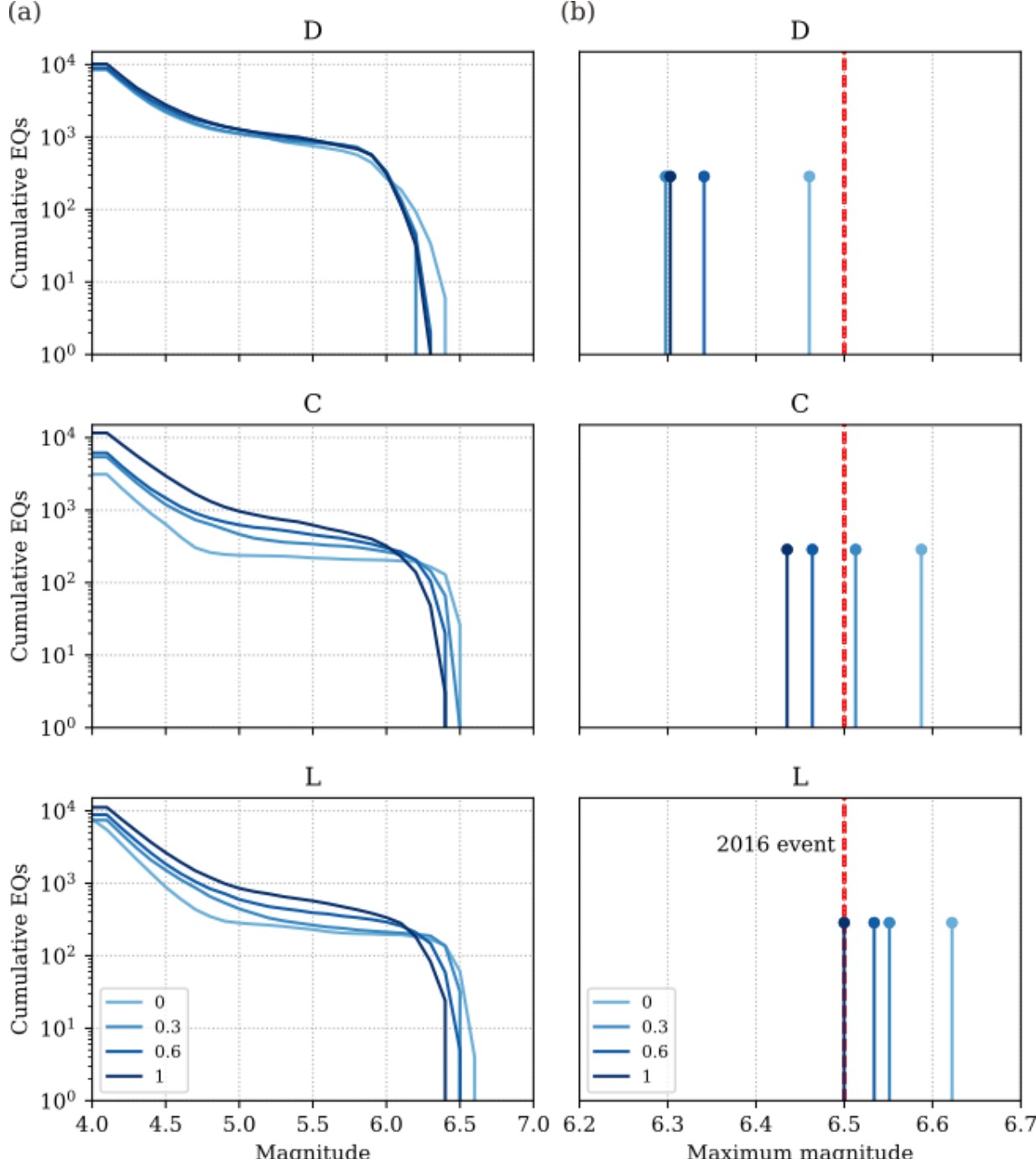

**Figure 7. a) Magnitude-frequency distributions (MFD) and b) maximum magnitudes of each one of the models grouped by their depth connectivity level and dip. Dashed red line in panel (b) corresponds to the magnitude of the largest event of the M$_w$ 6.5 2016 Central Italy sequence.**

### 3.2. Probability of surface rupture

### 3.2.1. Regressions of surface rupture probability for the whole fault

Earthquake surface rupture probabilities are magnitude-dependent, increasing accordingly with increasing magnitude. However, the probability of surface rupture is independent of the number of earthquakes occurred in the analyzed period. Figure 8 shows the variation in total earthquake rate (i.e., the sum of earthquake rates across all magnitude bins) over equal-length time windows along the catalogue, as a function of earthquake surface rupture probability. The figure demonstrates that periods with higher earthquake productivity of a given magnitude—meaning a higher number of earthquakes—do not necessarily correspond to higher surface rupture probabilities in that magnitude bin. In other words, surface rupture probability is primarily controlled by earthquake magnitude, not by how many events occur in a given time window.

**Figure 8. Rate-independent probabilities of surface rupture for the two end-member geometric models. The catalogue is split into 18 kyr-long sub-catalogues resulting from moving 18 kyr-long time windows throughout the whole 50 kyr of the catalogue. Each sub-catalogue from a time-window is colored by the logarithm of the total earthquake rate ($M_w \geq 4.0$) for that whole time window.**

We fit logistic regressions to the magnitude-specific likelihood of surface rupture data points (see figure S5 for data points used to fit the regressions). The logistic regressions (Fig. 9) quantify the influence of fault connectivity and sinuosity on surface rupture probability. As shown in Table 2, the regressions for each model are expressed by the regression coefficients $a$ and $b$ (intercept and slope, respectively; not to be confused with the rate and state friction law coefficients), the coefficient of determination $r^2$ and the p-value indicating statistical significance.

The regression analysis shows that earthquake magnitude is a statistically significant predictor of probability of surface rupture across all fault geometric configurations explored (p-values $\leq 0.03$). Moreover, $r^2$ expresses the quality of the fits, which show good fit values ranging from 0.5 to 0.7 depending on the fault geometric model, but most frequently between 0.6 and 0.7.

| Model | Sinuosity | a | a error | b | b error | $r^2$ | p-value |
|-------|-----------|------|---------|-----|---------|-------|---------|
|       | 0         | -30.4 | 13.3 | 5.3 | 2.3 | 0.7 | 0.02 |
| D     | 0.3       | -25.5 | 11.1 | 4.6 | 2.0 | 0.7 | 0.02 |
|       | 0.6       | -25.8 | 11.5 | 4.6 | 2.1 | 0.7 | 0.03 |
|       | 1         | -19.6 | 8.4  | 3.5 | 1.5 | 0.6 | 0.02 |
|       | 0         | -22.0 | 9.5  | 3.7 | 1.6 | 0.7 | 0.02 |
| C     | 0.3       | -15.3 | 5.9  | 2.8 | 1.1 | 0.5 | 0.01 |
|       | 0.6       | -19.7 | 7.6  | 3.6 | 1.4 | 0.6 | 0.01 |
|       | 1         | -28.2 | 12.1 | 5.1 | 2.2 | 0.7 | 0.02 |
|       | 0         | -19.7 | 8.0  | 3.4 | 1.4 | 0.6 | 0.02 |
| L     | 0.3       | -22.2 | 8.6  | 4.0 | 1.5 | 0.7 | 0.01 |
|       | 0.6       | -21.5 | 8.5  | 3.8 | 1.5 | 0.6 | 0.01 |
|       | 1         | -31.3 | 13.7 | 5.5 | 2.4 | 0.7 | 0.02 |

**Table 2. Logistic regression coefficients a and b, respective errors, coefficient of determination $r^2$ and p-value of each model explored in the study.**

For all models, surface rupture probability regressions (Fig. 9) show a sharp increase between $M_w$ 5.0 and $M_w$ 6.0. Above $M_w$ 6.0, nearly all ruptures reach the surface, while below $M_w$ 5.0, fewer than 10–20% do.

Surface rupture probabilities are largely impacted by geometry, both in terms of fault connectivity at depth and sinuosity. Disconnected models generally give more negative intercepts and higher slopes ($a$ and $b$ coefficients of the regressions, not to be confused with Gutenberg-Richter distribution parameters), which means that surface ruptures in these configurations require larger magnitudes. Connected models, especially those with constant dip (C), generally show less negative intercepts and lower slopes, indicating that connectivity facilitates lower magnitudes to rupture the surface.

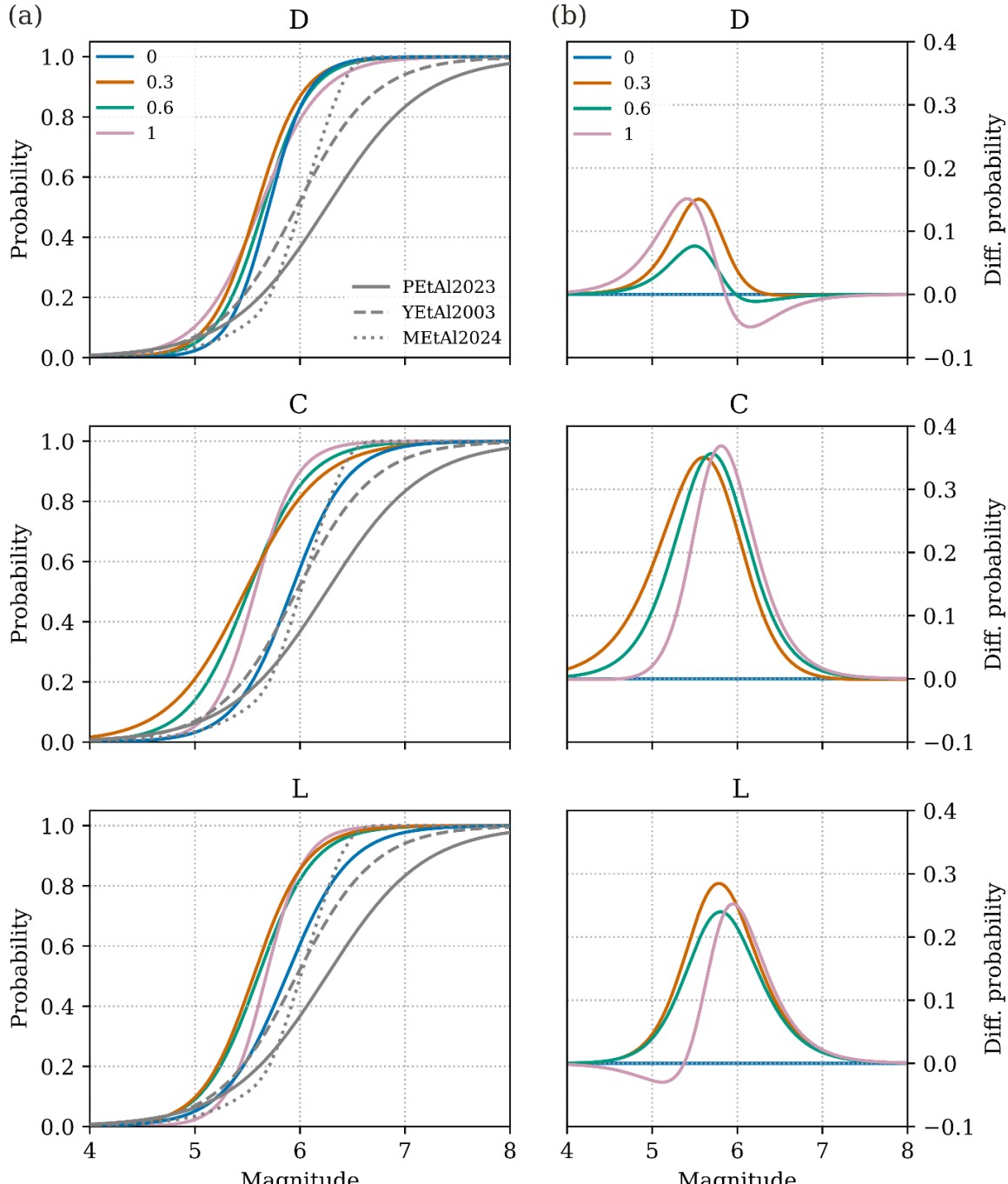

**Figure 9. a) Regressions of surface rupture probability for the different sinuosity models grouped by their depth-**
355 **connectivity level and dip. The simulated regressions are compared with the empirical regressions for normal faults**
**from Pizza et al. (2023) (PEtAl2023) and Youngs et al. (2003) (YEtAl2003), and with the numerical regressions from**

**Mammarella et al. (2024) (MEtAl2024). b) Residuals (differences) between regressions of panel (a) taking the sinuosity level 0 models as reference, i.e., each line is the result of subtracting the level 0 model from the corresponding sinuosity model.**

360 Fault sinuosity also plays a key role in surface rupture probabilities. For a same connectivity level, the models without fault trace sinuosity (i.e., level 0) consistently return lower surface rupture probabilities for the range of $M_w$ 5-6 compared to the models with sinuosity (Fig. 9). This observation is especially apparent in the connected models, where the surface rupture probability differences between sinuosity end members can exceed 0.3 in the $M_w$ 5-6 range (Fig. 9b). Increasing fault sinuosity also results in steeper regression slopes (*b* coefficients; table 2) in the connected models, thus shifting and narrowing the range

365 affected by these differences towards larger magnitudes. For instance, in the C group, the C1 model yields higher slopes than the C0 (*b* coefficients) increasing from 3.7 to 5.1, respectively. Contrastingly, the disconnected models show smaller differences regardless of their sinuosity, with maximum variation in predicted probabilities between sinuosity end members around 0.1 (Fig. 9b). This suggests that fault sinuosity has a secondary role when segments are disconnected at depth.

When comparing equal sinuosity levels (Fig. S6 in the Supplement), we identify that considering fault trace sinuosity, always

370 returns higher probabilities in the range of $M_w$ 5-6 (0.1-0.2 differences in probability) for the C model. Conversely, when removing sinuosity (level 0), the D model returns higher probabilities (differences up to 0.3 in probability), while C and L models produce nearly identical regressions (differences <0.05 in probability). This behavior is reflected in the logistic coefficients, with the D0 model being the one with the highest slope ($b$=5.34), while in the connected counterparts the highest slopes are for the sinuosity level 1 members (e.g. $b$= 5.5 in the L model). Moreover, introducing fault dip variability slightly

375 dampens probability of surface rupture for the $M_w$ 5-6 range in the models with sinuosity. This results in probability reductions up to 0.15 and by a shift in the regressions towards larger magnitudes (Fig. S6), as shown by more negative *a* regression coefficients (Table 2).

Hypocenter depth, governed by the geometric assumptions of the models, is the principal cause for changes in surface rupture probabilities. In the connected models, introducing fault trace sinuosity increases the density of shallow nucleations of larger-

380 magnitude events (around $M_w$ 6), which explains the higher surface rupture probabilities and steeper logistic slopes as sinuosity increases (Fig. 6). Conversely, lower sinuosity models nucleate smaller magnitudes in shallower locations, which results in flatter logistic slopes and more gradual increases in probability. In the disconnected models, the relative invariability in the hypocenter depth distributions explains the convergence in their regressions regardless of their geometry. In addition, introducing fault dip variability at depth slightly reduces the density of shallower nucleations, which shifts the regressions

385 slightly towards larger magnitudes; that is, surface rupture becomes more likely only for larger magnitude events.

In summary, the larger impacts in the regressions are related to the consideration of fault trace sinuosity (especially the one at surface) and with the linkage of fault segments at depth together. Considering both parameters consistently returns higher surface rupture probabilities for the $M_w$ 5-6 range and steeper regression slopes compared to considering just one. In contrast,

removing surface trace sinuosity or introducing listric geometry in connected models reduces surface rupture probabilities, while varying sinuosity of the fault at the base of the seismogenic depth has a limited impact on these probabilities.

### 3.2.2. Space-variable surface rupture probabilities

Figure 10 shows the surface rupture probabilities along strike for all the models explored in this study and for $M_w \geq 6$ events; those with higher chance of surface rupture according to the regressions (Fig. 9). The geometric connectivity of the faults at depth has the largest control on the spatial distribution of the surface rupture probabilities along fault (Fig. 10). Assuming that fault segments are disconnected at depth results in lower spatial rupture probabilities compared to the connected models.

Disconnected, D, models show peak probabilities from 0.45 to 0.52, depending on the sinuosity. In contrast, connected models show probabilities ranging from 0.67 to 0.88, and from 0.62 to 0.91 for the C and L models, respectively.

The disconnected assumption implies a large control of the segmentation on the spatial variability of the surface rupture probabilities. This configuration generates regions along the fault, like the Mt. Porche segment (Fig. 10) where probabilities are close to zero. This indicates that the disconnected geometry does not favor the generation or propagation of $M_w \geq 6.0$ in this segment. Conversely, the connected models produce higher surface rupture probabilities and show an along-strike distribution that tapers toward the fault tips. This pattern mirrors the slip rate distribution (Fig. 3), which is not observed for the disconnected model.

Regarding the effect of sinuosity, this parameter has little influence on the spatial distribution of the probability in the disconnected models. However, in the connected models the relationship between surface trace and bottom trace significantly impacts this distribution. Among the C and L scenarios, the sinuosity level 0.3 configuration yields the highest spatial probabilities. This is attributed to a higher proportion of shallow nucleations (<7km; peak of hypocenter depth) of earthquakes $M_w \geq 6.0$ in the level 0.3 models (Fig. 6). These shallow nucleations might be related to the fact that, in this fault model, the sinuosity increases sharply from the linking depth of the segments (at 7km) toward the surface. Conversely, the sinuosity levels 0.6 and 1 show more gradual sinuosity changes, which might reduce the generation of shallow large magnitude nucleations, thus lower spatial probabilities (Fig. 10).

While the influence of sinuosity on spatial probabilities differs from the trends observed in the magnitude-dependent regressions described in section 3.2.1, both approaches provide complementary insights. Spatial probabilities aggregate all events with magnitudes equal to or above the given threshold ($M_w \geq 6$), offering an estimate of where a large event is more likely to rupture. Conversely, regressions are evaluated for discrete magnitude bins and indicate how surface rupture probabilities scale with individual event magnitudes. This means that, unlike the regressions, the spatial probabilities not only correlate with earthquake nucleation depth but also with the magnitude range above the threshold captured by the catalogue (i.e., the number of earthquakes with magnitudes above that threshold). Therefore, for models with similar predominant nucleation depths (i.e., sinuosity levels 0.3 to 1 in both C and L configurations) those with larger $M_{max}$ (i.e., lower sinuosity) will show higher spatial probabilities.

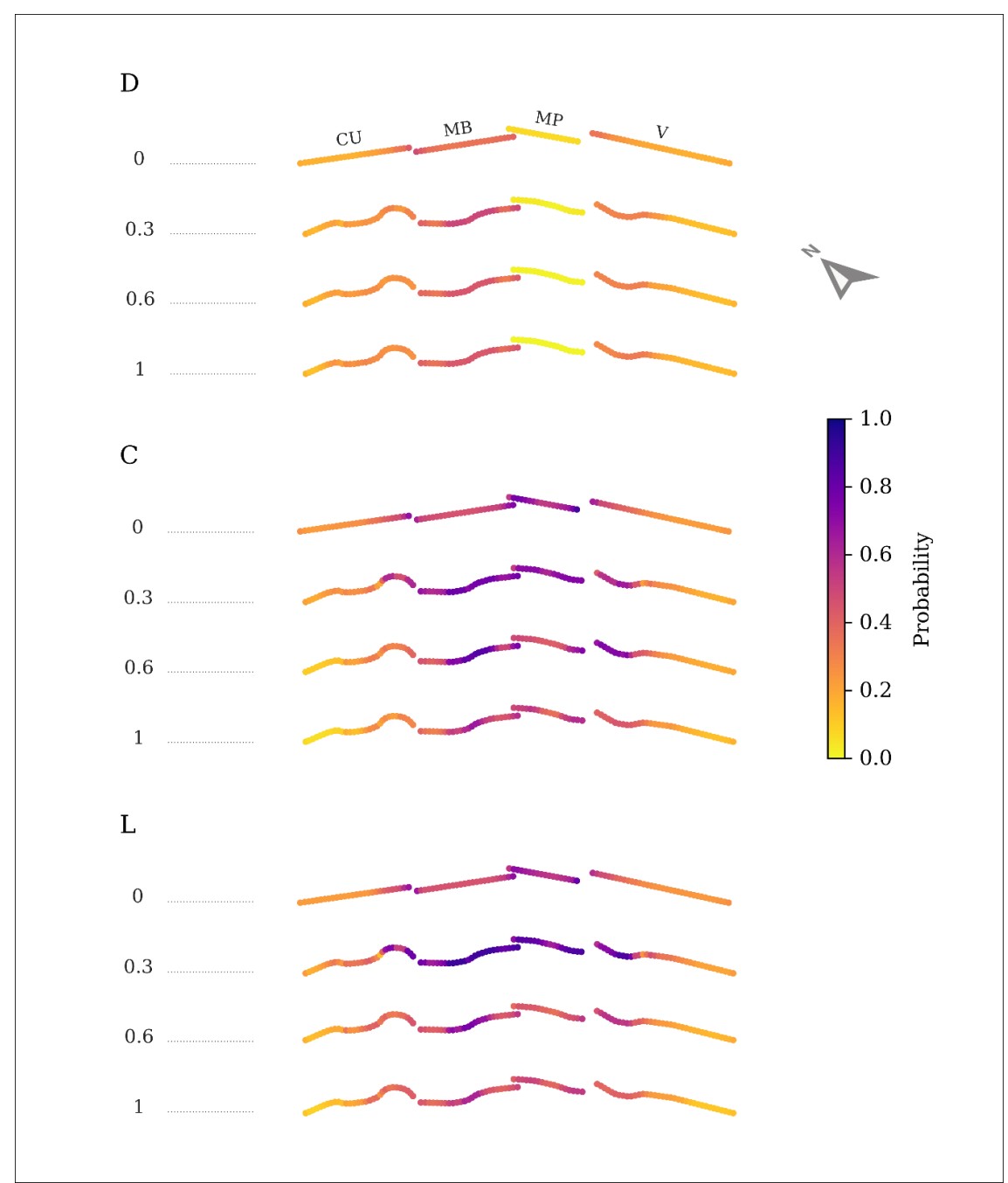

**Figure 10. Along-strike surface rupture probabilities for earthquakes of M_w ≥ 6.0 for the different geometric configurations explored in this study. Each row shows the surface fault trace of each model colored by their rupture probability. The geological fault segments (see Fig. 1) are indicated in the first model for reference. CU: Cupi-Ussita; MB: Mt. Bove; MP: Mt. Porche; V: Vettoretto.**

## 4 Discussion

In this section we discuss the implications of considering different geometric fault models into the simulated statistics of the earthquake catalogues. We also discuss the agreement of our simulations with observations as well as their methodological applicabaility to PFDHA.

### 4.1. Impact of fault geometry on the earthquake catalogues

Fault geometry plays a key role on the characteristics of the resulting simulated earthquake catalogues. The depth connectivity assumptions primarily influence the earthquake nucleation depth; while disconnected models nucleate earthquakes primarily following the depth-variable fault slip rate distribution prescribed to the models, the connected models primarily nucleate earthquakes at the depth where the fault segments merge. The sinuosity, on the other hand, favors shallower larger magnitude earthquake nucleations in models with depth connectivity.

Delogkos et al. (2023) recently investigated the impact of variable fault geometries and slip rates on simulated RSQSim earthquake catalogues and found outcomes consistent with those in our study. For one, they show how pre-imposing variable slip rate distributions, tapered toward the fault edges, enables a more realistic modeling of hypocenter depth distribution. For another, they demonstrate that increasing fault complexity (e.g., through the inclusion of antithetic structures or fault trace sinuosity) not only promotes shallower earthquake nucleations but also results in a more distributed hypocenter pattern across depth, rather than a single dominant nucleation level. Our models exhibit these same features. In addition, the depth distributions of more complex models (i.e., connected and with sinuosity) show better agreement with empirical distributions from Mammarella et al. (2024), compared to simpler configurations (Fig. 6).

In terms of earthquake catalogue statistics, we observe that the explored geometric features imply large differences in $M_{max}$ and MFD shape. Higher connectivity generally increases the $M_{max}$ due to the larger available rupture areas and a reduction in rupture segmentation barriers. In this context, there are models, especially the disconnected ones, that fail to reproduce magnitudes observed in the system, such as the Mt Vettore 2016 $M_w$ 6.5 earthquake. This discrepancy indicates that the disconnected models underperform in terms of $M_{max}$ compared to their connected counterparts, and could serve as a criterion for assessing fault model plausibility in the region.

Sinuosity, on the other hand, tends to reduce the $M_{max}$ by acting as a frictional barrier that can inhibit slip and preclude full growth of large earthquakes over the entire fault (Dieterich and Richards-Dinger, 2010). This explains why smooth faults yield characteristic MFDs, while rough or geometrically more complex faults produce MFDs closer to a Gutenberg-Richter (e.g., Delogkos et al., 2023). Dieterich and Richards-Dinger (2010) already observed that fault roughness progressively shifts the characteristic peak to shorter inter-event times and increases the rates in the magnitude gap between the power-law and the characteristic domains of the MFD. However, fault roughness alone does not fully overcome the tendency towards characteristic behavior, as we also observe in our results. According to the authors, other model parameters such as fractal

segmentation have a stronger impact on aiding this earthquake deficiency, and therefore is a feature that should be explored in future investigations.

## 4.2. Geometric controls on surface rupture probabilities

One of the key outcomes of our analysis is that fault geometric assumptions are the primary control on surface rupture
probabilities. Moreover, our findings are transferrable beyond the modeled case because they are independent of the earthquake rate along the catalogues.

Hypocenter depth is the key driver of surface rupture probability changes in our models. This observation is consistent with the conclusions on the numerical approach introduced by Mammarella et al. (2024). The authors identified the seismogenic depth as the main factor controlling the surface rupture probabilities, which controls the hypocenter depth. In line with this,
the relatively short seismogenic depth in our models, combined with stable nucleation depths, explain why regressions show sharp increases in probability in a relatively narrow magnitude range between $M_w$ 5 and 6. Shorter seismogenic depths nucleate shallower earthquakes, favoring the saturation of the seismogenic layer faster and thus allowing ruptures of a same magnitude to reach the surface easier than for deeper seismogenic depths.

Another important factor regarding nucleation is the location relative to the fault plane. The slip rate distribution, tapered to
470 the whole fault, generates slip rate concentrations in the fault segment edges that lead to a significant number of earthquake nucleations in these regions (see figure S7), a phenomena that has been widely described in simulators (Shaw, 2019). However, the impact of these earthquakes is minimal in the surface rupture probabilities. First, most of these nucleations correspond to low magnitude earthquakes ($M_w$ 4-5), which nucleate at consistent depths with depth distributions in the region (see figure S4) and rarely generate surface ruptures (Fig. 9a). For larger magnitudes $M_w \geq 5.0$, these anomalous nucleations are significantly
reduced (Fig. S7). Second, even though local probability increases at segment boundaries (Fig. 10) could be linked to the few $M_w \geq 6.0$ earthquake nucleation at fault edges, these can also be explained by geometric complexities, such as fault bends and segment connection at depth. Such geometric controls on nucleation have been previously described in the Mt. Vettore fault. For instance, Lavecchia et al. (2016) documents that the 2016 Mw 6.0 Amatrice earthquake nucleated at an inter-segment zone, where two fault segments link at depth. Third, even if surface rupture probabilities along-strike are slightly affected by
anomalous earthquake nucleation, in a full PFDHA application the fault displacement hazard is ultimately driven by the slip recorded in each site. Our models show that both coseismic and cumulative throw taper toward segment edges (Figs. 11 and 12) independently of nucleation location, suggesting that localized increases of surface rupture probability are unlikely to bias fault displacement hazard.

Fault geometry, especially trace sinuosity and segmentation, also has a critical impact on earthquake rupture behavior.
Geometric roughness acts as a physical barrier that affects how ruptures propagate (e.g., Dieterich and Richards-Dinger, 2010; Zielke and Mai, 2025). For instance, introducing fault trace sinuosity in our models hinders lateral rupture propagation and instead favors along-dip rupture, which raises the probability of surface rupture in connected configurations. Likewise, fault

connectivity favors segment-to-segment rupture propagation. When combined with no sinuosity (i.e., level 0) surface rupture probabilities are reduced as ruptures are able to propagate laterally, which is not observed for the disconnected counterparts that still preserve lateral geometric barriers (segmentation). Similarly, listric fault geometries generally reduce the probabilities of surface rupture in the regressions because the fault area available is larger, requiring larger magnitudes to reach the surface.

These modeling results are consistent with numerous field observations indicating that fault surface geometry strongly controls rupture propagation and slip patterns (e.g., Lettis, 2002; Rockwell and Klinger, 2013; Rodriguez Padilla et al., 2024). For example, the recent study by Rodriguez Padilla et al. (2024) showed that geometrical features, such as step-over width, are key locations for rupture arrest. This is consistent with the observed drop in surface rupture probabilities of $M_w \geq 6.0$ earthquakes across the step-over between the Mt. Porche and the Mt. Bove segments (Fig. 10). This agreement between simulation and observations reinforces the reliability of earthquake simulators for characterizing fault rupture behavior at the surface. In addition, earthquake simulators offer the opportunity to systematically quantify the influence of fault geometric features on rupture propagation, such as earthquake jump distance as a function of the fault slip. This is an issue that might be investigated in the future.

Our results also indicate that surface fault trace geometry has a stronger influence on magnitude-dependent rupture probability than the subsurface fault trace. In this sense, constraining fault traces at surface should be prioritized over subsurface traces. This finding supports recent conclusions by Zielke and Mai (2025), who demonstrated how models with a same surface fault trace produce long-term fault behavior results that are interchangeable, even if their subsurface geometry differs significantly. That said, fault segment connectivity at depth, especially when paired with surface fault trace sinuosity, remains a dominant factor controlling surface rupture probabilities in our models and therefore should not be neglected.

The strong influence of fault geometry in earthquake surface rupture statistics contrasts with the frequent lack of constraints on the subsurface geometric features of faults, often very expensive and difficult to image. In addition, frequent challenges in the identification of primary fault traces and uncertainties in fault trace location can also become a limitation for the correct characterization of fault geometries and, thus, for the implementation of the proposed approach. To tackle these issues in a hazard evaluation context, exploratory analyses represent the most suitable approach. For instance, as noted by Zielke and Mai (2025), exploring multiple realizations of fault geometries, including fault trace hypotheses, may be a practical solution to account for the epistemic uncertainties linked to the poor knowledge of subsurface fault geometries.

### 4.3. Comparison with empirical and numerical regressions

Our surface rupture probability regressions are generally closer to those derived from numerical approaches (Mammarella et al., 2024) than those from empirical earthquake data (Pizza et al., 2023; Youngs et al., 2003). On the one hand, simulation and numerical approaches generally show steeper slopes than the empirical regressions due to fault-specific modeling assumptions. That is, in both approximations the models are constrained to a single seismogenic depth (12 km), fault dimensions, and similar

hypocenter depth distribution. This is further corroborated by the higher coincidence between the C0 and L0 regressions and the numerical regressions. The geometric considerations of the sinuosity level 0 models are closer to the ones by Mammarella et al. (2024), where fault sinuosity is not considered. Interestingly, such a coincidence between regressions is not observed for the D0 model probably because the strict fault segmentation limits earthquake lateral growth.

On the other hand, empirical regressions are based on mixed datasets of global earthquakes occurring on faults with different seismogenic depths, hypocentral distributions and geometries, which may smooth out the regressions. Modeling several fault systems would return regressions closer to the empirical ones, as they would result from a broader range of tectonic settings and parameters that do not change so much in a single fault system. To prove this, we tested how mixing fault models with different geometries and seismogenic depths influences the regressions (Fig. S8). We combine the catalogues from six different geometric models into a unified dataset to better approximate large-scale (multi-fault system) analyses in PFDHA. These models are selected to capture the broadest range of variability in regression behavior (Fig. 9), and we include two new models with seismogenic depths that differ from those assumed in our study. This analysis shows how mixing fault models with different seismogenic depth and geometric considerations generally smoothes out the regression slopes and shifts them toward larger magnitudes, similar to what is observed for empirical regressions (Fig. S8).

Generally speaking, the regressions of surface rupture we obtain are visibly off the empirical and numerical regressions available in literature. There are several modeling factors and assumptions that may contribute to this effect.

First, the geometric components explored in this study are a sample of the whole spectrum of potential geometrical complexities to be combined and explored. Here we focused on fault connectivity and trace-scale complexities, but smaller scale variability might also be explored. For instance, Zielke and Mai (2016) demonstrated how incorporating sub-patch geometrical roughness – i.e., roughness at spatial scale below fault element size – impacts earthquake behavior in earthquake rupture simulations. Similarly, exploring the impact of fractal roughness as in Allam et al. (2019) might provide new insights into fault surface rupture behavior.

Second, the model parameters, namely the uniformity in the initial stresses and rate-and-state coefficients (*a-b*), are considerable simplifications. Initial stresses are uniformly distributed throughout all elements in the simulations, while in reality stresses change at depth (e.g., normal stress increases as a function of depth and is also dependent on fluid pore pressure). This implies that stress conditions equivalent to 6 km depth are assigned to all fault elements, including near surface. Even though in RSQSim the initial stress conditions evolve throughout the simulation, these initial conditions affect how ruptures propagate (e.g., Liao et al., 2024), thus how they manifest at the surface.

We assumed uniform stresses to prevent shallow-dominated hypocenter nucleations caused by small stress values near surface in heterogeneous stress models. As identified by Liao et al. (2024), heterogeneous stresses would shift the hypocenter nucleations to unrealistically shallow depths, ultimately increasing the probability of surface rupture for all models. Even though depth-variable stresses might yield more realistic fault slip distributions, we decided to prioritize accuracy in nucleation

depth given its impact in our analysis. Building on this idea, Hughes et al. (2024) used uniform initial stresses for RSQSim-based tsunami models to avoid shallow nucleation depths of heterogeneous stress models and to ensure more conservative (higher) wave heights for the hazard assessment, even if their earthquake slip patterns are less realistic.

Similarly to the initial stresses, RSF law coefficients a and b have been assumed uniform for all fault elements with velocity weakening conditions ($a\text{-}b<0$). In nature, however, these coefficients follow depth dependent profiles, with typical regions near surface under velocity strengthening conditions ($a\text{-}b>0$; e.g., Lapusta et al., 2000). While introducing such variability can improve some catalogue features like the depth distribution of earthquakes (especially when paired with depth-variable stresses; Liao et al., 2024) and rupture propagation, it would also add complexity to our analysis. Data on the depth variability of RSF parameters is generally not available in most regions, which would likely require adopting unvalidated assumptions in the model parametrization.

The roles of stress and frictional parameters are definitely topics to be explored in the future given the demonstrated impact they have shown in simulated catalogues (e.g., Delogkos et al., 2023; Gómez-Novell et al., 2025a; Liao et al., 2024). As an example, we have tested the influence of varying ($a\text{-}b$) in the regressions of our end-member models used for the benchmarking tests (see Table 1). We observe that these parameters have a strong influence in the probabilities of surface rupture (Fig. S9 in the Supplement), but not so much in the shapes of the regressions. In general, more negative ($a\text{-}b$) decrease the probabilities of surface rupture by shifting the regressions toward larger magnitudes. In addition, the regression variability linked to the ($a\text{-}b$) variations depends on the fault geometric model; higher in the L geometry compared to the D (Fig. S9). This lower sensitivity to geometric changes in the disconnected model is consistent with the observations made in section 3.2. Although the ($a\text{-}b$) parameters used in our study are the ones that better match magnitude-area empirical relations, we underscore the importance of constraining these physical parameters where earthquake simulation studies are conducted.

Despite the modeling factors, it is important to remark that part of the misfit between simulated and empirical regressions may also be linked to observational biases affecting empirical data. That is, surface ruptures with short rupture lengths, small displacements or those occurred in remote regions or historical times are more likely to be underreported or missing in databases. These omissions can ultimately lead to underestimated empirical regressions. Other causes for lower surface rupture probabilities in empirical models can be related to near-surface soil conditions. For instance, the presence of soft sediments, uncompressed rocks or loose materials can lead to fault offset attenuation and accommodation through warping or folding, ultimately decreasing the imprint or even recognition capabilities of surface ruptures.

A final limitation is that out study focuses solely on on-fault surface displacements, while distributed rupturing and off-fault deformation are important components of empirical PFDHA models. While the implementation of such components is beyond our scope, our work can serve as a basis for future work in this direction. For instance, the simulated surface displacements on the principal fault can be combined with empirical distributed faulting regressions (e.g., Visini et al., 2025) to develop

probabilistic models of distributed fault ruptures. An example of that is the work by Daglish et al. (2025), where they use the simulated on-fault displacements to scale across-fault displacement based on empirical data on surface rupturing earthquakes.

## 4.4. Suitability of the models based on observations

We compare how well our models reproduce surface geological observations of the fault to test the suitability of our model parameters and future applicability to PFDHA.

### 4.4.1. Coseismic ruptures

We compare the modeled along-strike coseismic surface slip distributions of large magnitude events with the observed coseismic slip of the 2016 $M_w$ 6.5 Mt. Vettore earthquake (Fig. 11). Overall, the simulated coseismic slip values agree with
observations along most of the fault, except in the southern sector of Mt. Vettore. In this region, the simulated models systematically underestimate the slip recorded during the 2016 event, which was described as an anomalously high slip by several authors (e.g., Brozzetti et al., 2019; Puliti et al., 2020; Villani et al., 2018). The causes for this high surface slip have been attributed to local fault geometric features such as persistent fault dip irregularities at depth (Brozzetti et al., 2019), all features that we did not consider in our modeling. Other authors have considered also the influence of gravitational processes
on the total slip recorded in this sector of the fault (e.g., Di Naccio et al., 2019), a phenomena that is not considered in earthquake cycle modeling.

Fault trace sinuosity at surface clearly improves agreement with the 2016 cosesmic slip observations, reducing the mean absolute error (MAE) by 30-40% (Fig. 11). Sinuosity generates geometric barriers along fault surfaces that likely attenuate stress transfer and rupture growth, decreasing earthquake slip at the surface in comparison to the smoother fault surfaces.
Another relevant observation is that sinuosity generates higher spatial variability – i.e., along-strike fluctuations – in the coseismic slip distributions compared to the smoother fault models. However, the simulated cosesmic distributions event to event show overall less dispersion, higher convergence and higher predictability in all models with fault trace sinuosity, regardless of the depth geometric assumptions (Fig. 11). These findings are consistent with RSQSim simulations performed by Allam et al. (2019) on fractally rough faults, which established a quantitative link between fault fractal roughness and
earthquake properties such as increasing coseismic slip variability. All these features are consistent with observations on earthquake behavior and suggest that geometrically rougher surfaces might be more realistic in simulating coseismic events.

While the surface fault trace plays a crucial role in the coseismic slip response of the models, the impact of the trace at the base of the seismogenic depth is more modest, leaving MAE values practically invariable across models; i.e., MAE variations in the order of <10 cm slip (Fig. 11). We observe this behavior beyond the coseismic observations. For instance, the variations
in MFD shape and $M_{max}$, as well as the surface rupture regressions are less significant across models that share the same fault trace at surface than compared to those that consider a straight fault trace (without sinuosity). Such similarities in the

catalogues that share the same fault trace at surface but with varying geometry at depth is a model behavior that is again consistent with recent analyses made with the MCQsim earthquake cycle simulator by Zielke and Mai (2025).

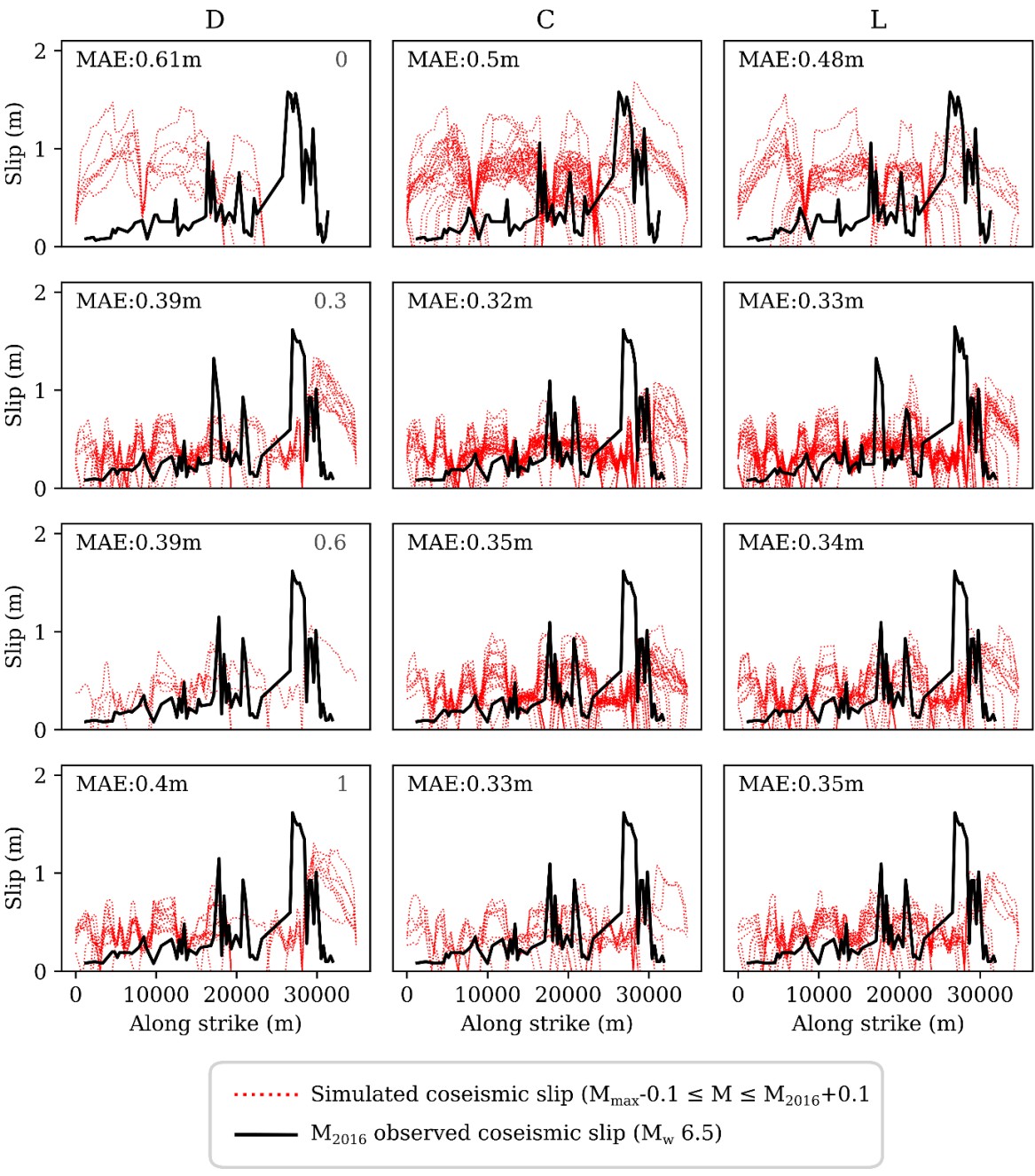

**Figure 11. Comparison between the observed surface coseismic slip along strike of the 2016 M$_w$ 6.5 Central Italy earthquake on the Mt. Vettore main fault (from Nurminen et al., 2022) and the stack of surface coseismic slip profiles**

**from simulated events of similar magnitude in the whole 50 kyr-long catalogue. Each column groups the results of models with the same connectivity and dip at depth and the rows group models by their sinuosity level. The agreement between observation and simulations is represented by the mean absolute error (MAE): the average of differences between the simulated slip values and the observed ones. Lower MAE means better agreement. Along strike distances are calculated from the NW to SE (see map in figure 1).**

Regarding the fault connectivity, we do not observe a significant impact on the agreement with coseismic observations. However, some models, especially the disconnected ones, fail to reproduce magnitudes in the order of the 2016 event, contrary to the connected counterparts (Fig. 7b). This issue inherently limits the performance of the disconnected models in reproducing coseismic behavior.

Our findings on coseismic rupture behavior have important implications for both seismic hazard and, in particular, PFDHA. The common practice of simplifying fault traces for modeling purposes – removing fault geometric roughness – might lead to less realistic simulations of surface displacement. This limitation is especially relevant for fault displacement forecasts in PFDHA, where capturing the spatial behavior of the surface ruptures is key.

As discussed earlier, introducing local scale fault geometric complexities could help aid geometric discrepancies observed in the coseismic slip distributions at the southern end of the fault (Fig. 11). However, resolving more detailed geometries implies a much higher fault discretization level that directly translates into higher computational demands, while its translation to more accurate and realistic simulations is not necessarily straightforward.

### 4.4.2. Cumulative throw

The comparison between simulated and observed cumulative throw highlights several important insights regarding model performance and fault behavior.

In all models, the simulated cumulative throw of $M_w \geq 5.5$ events for 18 kyr time windows of the catalogue considerably overestimates the measured values for most of the fault trace, except for the southern sector, where it is underestimated (Fig. 12). The along-strike trend of the cumulative throw is comparable to the simulated one, especially in the inter-segment regions where both simulated and observed cumulative throw drop. This evidences that the simulations are able reproduce geometric slip patterns that are observed in these critical regions along fault. Importantly, these inter-segment regions control earthquake surface rupture probabilities, especially in disconnected geometric configurations. As for the coseismic slip, the large discrepancy in the southern tip of the fault this is likely due to geometric complexities of the fault at depth that have not been accounted in our models (see section 4.4.1).

The variability between the simulated and observed cumulative throw across models is significantly smaller than for the coseismic case, with MAE values that vary around 10-15% between geometric assumptions (Fig. 12). Even though the geometric assumptions of the different models slightly affect the fit to cumulative throw observations, there is no clear

correlation between model fit and geometric assumptions. In fact, cumulative throw is mainly correlated to the intial slip rate conditions of the model (Fig. 3), which are virtually equal across all geometric configurations. The tapered shape in the cumulative throw toward the fault edges replicating that of the slip rate evidences this correlation (Fig. 12). RSQsim employs the back-slip approch, in which the total amount of slip at the end of the simulation can be predicted by the product of the slip rate and the catalog length (assuming all slip is seismic).

Slight variations in the MAE come from geometric assumptions that: 1) might act as local barriers for slip, adding or removing variability in the cumulative throw curve accordingly, and 2) change the triangular mesh configuration at the surface, thus changing the field data points that are assigned to each fault element (see details in section 2.6).

Generally speaking, cumulative throw is a less reliable measurement to confront simulations with because it is subject to long-term geologic phenomena not related to tectonics, such as erosion or gravitational movements. Erosion is quite important in the Mt. Vettore region. For instance, the erosive phenomena in the Ussita Valley (northern sector; Fig. 1) obliterates geomorphic footprints of long-term fault activity as recognizable by the lack of cumulative throw data in the profile (e.g., Puliti et al., 2020). Gravitational phenomena such as landslides are also present in the high-mountain setting of the fault, which can severely modify tectonic deformation evidence (e.g., Di Naccio et al., 2019). Along this line of reasoning, a large part of the systematic overestimation from the simulated cumulative throw along fault can be explained by the absence of erosion correction in our models, a feature that is beyond the scope of this work.

Overall, although cumulative throw trends are partially captured, their value as a modeling constraint is limited due to long-term geomorphic modification and data gaps. Accounting for geomorphic – e.g., erosive –  corrections as well as increasing data sample points in future work would help widen the applicability of geomorphic markers to constrain earthquake simulations. Nonetheless, the impact of this issue is manageable in our study because matching single earthquake (coseismic) behavior is more relevant for PFDHA than the long-term cumulative one.

### 4.5. Insights on applicability of earthquake simulators to PFDHA

This study demonstrates that earthquake cycle simulators like RSQSim are a valuable tool for advancing PFDHA, in line with Daglish et al. (2025). While the current models are implemented for a specific fault system, our methodological approach provides insights that are transferable to PFDHA applications in general.

First, our simulations produce coseismic slip patterns that are reasonably coherent with observed data on the 2016 $M_w$ 6.5 Mt. Vettore earthquake.

Second, even though we have omitted long-term geomorphic processes and structural complexities at depth, the cumulative throw is partially captured by our models and its spatial trends are preserved with respect to observations (e.g., segment limits).

Third, the hypocenter depth distributions modeled match the regional observations in Italy and demonstrate its controlling role in the probability of surface rupture, findings that are in agreement with previous numerical approximations.

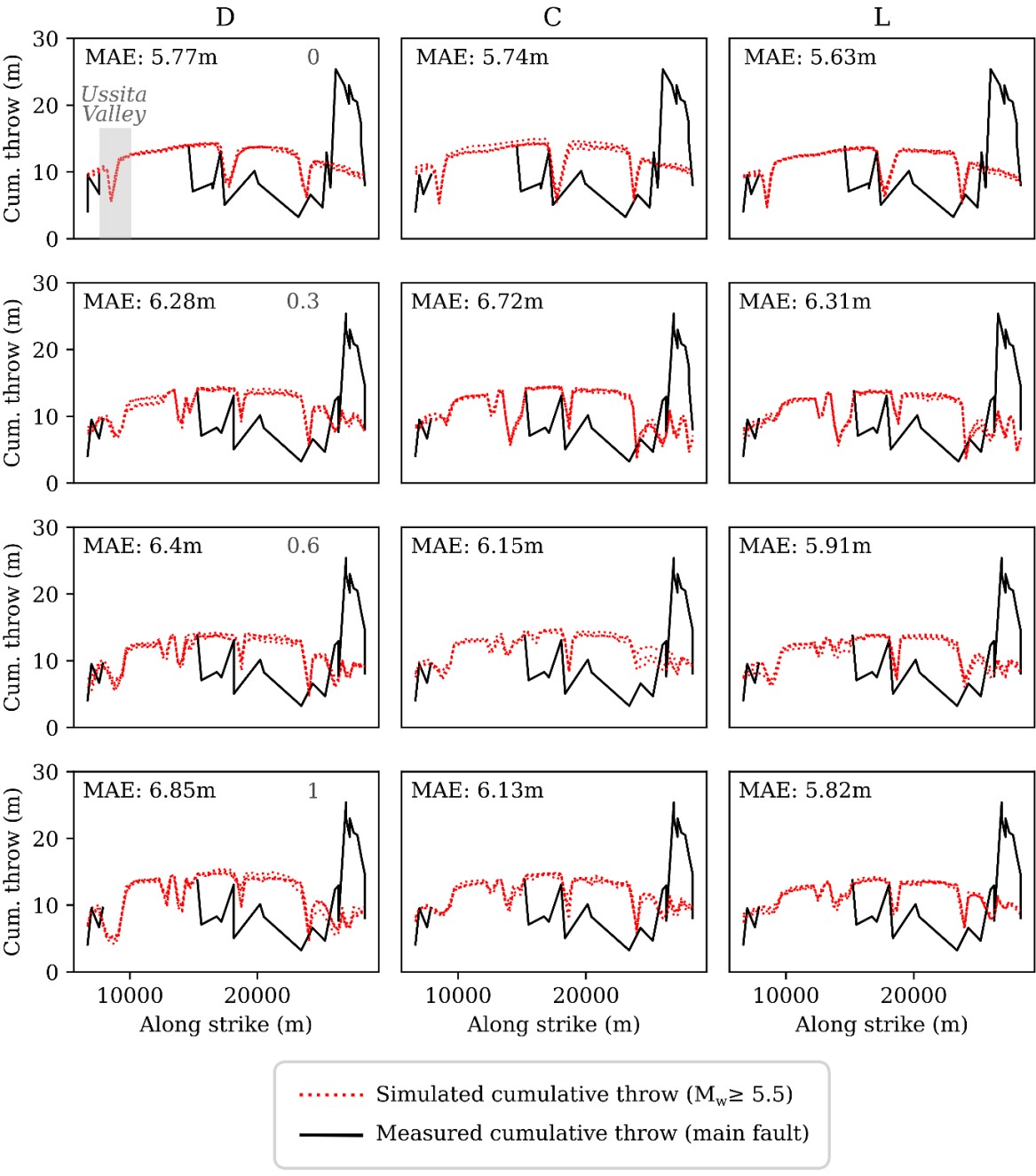

**Figure 12. Comparison between the measured cumulative throw along the main fault of the Mt. Vettore fault system**
**for the past 18 kyr (from Puliti et al., 2020) and the simulated cumulative throw ($M_w \geq 5.5$) for 18 kyr time windows**

**along the whole 50 kyr catalogue. Each column groups the results of models with a same connectivity and dip at depth and the rows group models by their sinuosity level. Like in figure 11, the agreement between observed and simulated values is expressed by the mean absolute error (MAE). The Ussita Valley location along strike is indicated in the first panel (upper left). See figure 1 for map location. Along strike distances are calculated from the NW to SE.**


Earthquake cycle simulators can also help to overcome inherent completeness issues of earthquake databases for fault displacement hazard, enabling enrichment of earthquake databases, the systematic exploration of fault parameters like subsurface geometry and allowing fault-specific analyses. By extension, the explicit consideration of earthquake rupture physics and the high-resolution earthquake displacement data generated by the simulations allows the implementation of site-

specific analysis with the displacement approach, one of the key challenges of PFDHA.

Consequently, as anticipated by Valentini et al. (2025b), simulation-based studies like the present one might contribute to the implementation of earthquake simulators into PFDHA in the future and to the overall enhancement of their capabilities. This enhancement potential has already been demonstrated for seismic hazard (e.g., Herrero-Barbero et al., 2023; Rafiei et al., 2022; Shaw et al., 2018) and tsunami hazard applications (e.g., Álvarez-Gómez et al., 2023; Hughes et al., 2024).

Despite the advantages, the use of earthquake simulators has limitations for PFDHA applications, some of which already described by Daglish et al. (2025). These include i) less extensive statistical validation with respect to empirically-based seismic hazard models, ii) rate-and-state parameters being calibrated to match magnitude-area scaling used for empirical approaches, preventing from a fully-independent analysis, and iii) important simplifications in the physics of earthquake rupture propagation, compared to fully dynamic rupture simulations. In addition, the general lack of site-specific data on the

fault systems, like in many numerical approaches, can become a challenge for the successful implementation and validation of the analyses proposed here. Having said that, to date these models offer the most computationally efficient solution to model earthquake cycles. These models provide a balance between reasonable earthquake rupture physics and the ability to generate near-surface displacements over several seismic cycles (e.g. Daglish et al., 2025), necessary for robust long-term fault statistics in PFDHA  In this line, the emergence of newer and improved earthquake cycle simulators such as MCQsim (Zielke and Mai,

2023), which allows the explicit incorporation of fault roughness and visco-elastic relaxation, or Tandem  (Uphoff et al., 2022), which enables fully-dynamic multi-cycle rate-and-state friction, could further enhance the capability of simulators to reproduce more realistic earthquake rupture sequences in the future. Additionally, when fault data is scarce, earthquake simulators offer the opportunity to systematically explore epistemic uncertainties in many fault model parameters, making it a strong alternative to fully empirically-based approaches in PFDHA.

**5 Conclusions**

In this study, we explore the influence of fault geometry on coseismic surface rupture probabilities using RSQSim earthquake cycle simulations at the Monte Vettore Fault System in Central Italy.

Our results evidence that fault geometry, specifically fault segment connectivity at depth and fault sinuosity, is a primary control on the probability of coseismic surface rupture.

Models with connected fault segments at depth increase surface rupture probabilities for the $M_w$ 5-6 range compared to their disconnected counterparts, particularly when combined with surface fault trace sinuosity (i.e., fault plane roughness driven by the fault traces) and constant dip. Both connectivity at depth and sinuosity promote shallower earthquake nucleation and favor rupture propagation toward the surface rather than laterally. In contrast, listric geometries and, especially, connected faults without sinuosity reduce probabilities of surface rupture due to greater available rupture area and reduced barriers for lateral
rupture propagation, respectively.

The depth distribution of earthquake hypocenters is the dominant physical parameter controlling surface rupture probability, which is driven by the geometric assumptions of the models. Variations in hypocenter depth strongly correlate with the ability of ruptures reaching the surface, a result that is consistent with recent numerical approaches. Fault segmentation assumptions also impact significantly on the spatial distribution of surface rupture probabilities in disconnected models, as they impose
physical limits on maximum magnitudes and therefore reduce surface rupture potential. These findings highlight the importance of accurately representing fault geometry in fault-specific displacement hazard assessments.

Comparisons with empirical and numerical surface rupture regressions reveal that our simulation approach matches numerical results better than empirical ones. This is due to the consistent seismogenic parameters and fault-specific setup in both simulation and numerical models, as opposed to the broader intrinsic dataset variability of empirical models.

Our simulation outputs also show strong agreement with geological observations in the region tested. Models generally agree with the observed coseismic slip of the 2016 $M_w$ 6.5 earthquake at the Monte Vettore, especially those that consider both depth connectivity and fault trace sinuosity. Cumulative throw patterns are less accurately reproduced due to long-term geomorphic processes not accounted for in our models. However, general spatial trends such as segment limits are consistent with field observations. These results support the validity of our approach to investigate site-specific surface rupture statistics for hazard
evaluation purposes.

Our findings demonstrate the potential of earthquake cycle simulators like RSQSim for improving probabilistic fault displacement hazard analysis (PFDHA), especially when empirical data are sparse or fault-specific assessments are needed. Our study indicates that physics-based simulators can strongly complement empirical regressions in PFDHA, particularly when used to investigate the spatial variability of surface rupture and the influence of fault geometric features in specific fault
systems.

Future work should explore the inclusion of depth-dependent stress conditions, variable frictional parameters, and finer-scale geometric complexities to further approximate real fault behavior. Extending this framework to other fault systems could help generalize our conclusions and support the development of next-generation PFDHA methodologies.

**Code availability**

The RSQSim code, developed by Richards-Dinger and Dieterich (2012), can be obtained from the authors upon request. All catalogue analyses, calculations and figures shown in this paper have been generated with Python 3.12.

**Data availability**

All the input files (i.e., RSQSim input parameter files and fault models) and outputs of all twelve simulated catalogues are available for download in a Zenodo repository under the CC-BY 4.0 license (Gómez-Novell et al., 2025b).

**Author contribution**

**OGN:** Conceptualization, Methodology, Software, Validation, Formal Analysis, Investigation, Data Curation, Writing (original Draft Preparation), Visualization; **FV:** Conceptualization, Methodology, Investigation, Writing (original draft preparation),**;** **JAAG:** Conceptualization, Methodology, Investigation, Resources, Funding acquisition, Writing (review and editing)**; BP:** Conceptualization, Methodology, Investigation, Funding acquisition, Writing (review and editing)**; JGM:**
Resources, Funding acquisition, Writing (review and editing).

**Competing interests**

The authors declare that they have no conflict of interest.

**Acknowledgements**

The authors are thankful to the Fault2SHA community for the fruitful discussions leading to the conceptualization and
development of this work.

**Funding**

This research work was funded by the European Commission – NextGenerationEU, through Momentum CSIC Programme: Develop Your Digital Talent (project GDxGT; ref. MMT24-IGME-02), by the Model_SHaKER project (ref. PID2021-124155NB-C31) funded by Spanish State Research Agency, and by the TREAD project (daTa and pRocesses in sEismic
hAzarD), funded by the European Union under the Horizon Europe programme: Marie Skłodowska-Curie Actions (MSCA) with grant agreement no. 101072699.

Octavi Gómez-Novell is staff hired under the Generation D initiative, promoted by Red.es, an organisation attached to the Ministry for Digital Transformation and the Civil Service, for the attraction and retention of talent through grants and training contracts, financed by the Recovery, Transformation and Resilience Plan through the European Union's Next Generation funds.

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
