# Peer review of "Coseismic Surface Rupture Probabilities from Earthquake Cycle Simulations: Influence of Fault Geometry"

_EGUsphere, 2025_

## Author Comment (AC2)

**Responses to Referee #2**

In the following document we provide detailed responses to the different comments and explanations on how we will implement this in a revised version of the manuscript.

The probability that an earthquake becomes a surface-rupturing event is a key ingredient in probabilistic displacement hazard analysis. Robust estimates of these probabilities are limited by the scarcity of surface rupturing events. Understanding how different fault properties, such as geometry, connectivity at depth, or sinuosity, affect this probability, is hampered by the lack of detailed observations at depth. Gomez-Novell et al. bring an innovative approach to this data gap. They use rupture simulators to test the effect of different fault geometries at depth on the probability that an event becomes a surface rupturing one on the Mt Vettore fault in Italy. Their study highlights how geometry influences the probability of surface rupture and offers a pathway to incorporate inferences from simulators into PFDHA. The contribution is original and useful and I support eventual publication.

Author's response (AR): We kindly thank the reviewer for the very positive feedback on the article.

I have some minor comments that are mostly focused on improving the clarity of the article:

Figure 2: the segmentation and smoothness degrees the authors test are very reasonable but I find the trace-trace trace-smooth etc. wording to be very confusing. I think the suite of geometries may be captured by two constraints: a segmentation (n of separate segments) constraint, and a roughness (for example, RMS roughness as used in fault roughness studies). These would describe the suite of geometries quantitatively and remove the confusion infused by the naming choices.

AR: This comment highlights a confusing selection on model nomenclature that we agree with and, as such, we will take appropriate measures to solve it in the revised manuscript. In detail, we will adopt a parametrized nomenclature. Letters for the connectivity level: D- Disconnected, C-Connected constant and L-Listric; increasing numeric values for increasing sinuosity/roughness of the fault, going from 0 (minimum sinuosity) to 1 (maximum sinuosity). As such, the models will become:

- Disconnected: D0 (Linear-linear), D0.3 (trace-linear), D0.6 (trace-smooth), D1 (trace-trace).
- Connected constant: C0, C0.3, C0.6, C1
- Listric: L0, L0.3, L0.6, L1

We believe this nomenclature for the sinuosity is more intuitive, helping readers to clearly understand the models that have higher or lower sinuosity, without the need to numerically compute it from the models. In the manuscript text (section 2.2), we will provide a proper introduction to this new nomenclature, and we will replace all mentions to the former nomenclature with the new one (figures and text).

Line 229 typo in extra them

AR: We will fix this in the revised manuscript.

Figure 7 - I don't understand why the connected listric would produce larger magnitude events than the connected constant - isn't the listric geometry quite unfavorable for slip propagating into those regions?

AR: The listric model produces larger magnitude events than the non-listric because the fault rupture area available is larger. Even though the listric geometry causes dip variations at depth, these dip variations are not very large and quite gradual. Consequently, the fault plane stresses likely do not show sharp transitions at depth enough to prevent ruptures from propagating.

Figure 8 - telling these models apart visually is a bit hard. Can the authors fit a logistic regression to highlight the differences between the two end-member models? Should be easy to do since the authors do it anyway to provide the parameters in the next table and have the regressions in Fig 9.

AR: The purpose of this figure is to show that earthquake rates for a given time period do not correlate with surface rupture probability, which is evident when looking at the magnitude bin-specific probabilities. While a logistic regression might improve the visualization, it would represent a fit to the data rather than the data itself. Such a fit can deviate from the actual values and potentially mask the rate-independence relationships we want to highlight.

Line 327 - the authors point out that a and b are not the rate and state friction coefficients but the intercept and slope of the logistic fits. This is a useful consideration. They should also point out that a and b are not the parameters in the magnitude-frequency distribution, since this is another possible source of confusion given the nature of the article.

AR: Agreed. We will add this clarification in the revised version in the manuscript.

Figure 10 - consider not using a divergent color map, since the probabilities go from 0 to 1.

AR: We will consider this for the revised manuscript. We will use a perceptually uniform sequential colormap like "plasma", as we show below.

I appreciate how this article weaves the modeling results with the results from empirical studies in the literature.

AR: We are thankful for the positive feedback on this comment.

The authors could refer to Valentini et al. (2025)'s call for more model-driven advances to supplement current PFDHA approaches as part of the justification for this work.

AR: We agree and we will add this in the revised manuscript version, specifically in section 4.5. Please note that Valentini et al. (2025)'s paper was not published when we submitted the manuscript.

---

## Author Response (AR1)

**Responses to Referee #1**

General comments:

The paper by Gómez-Novell et al investigates the role of fault geometry in the probability of surface rupture along a fault, using the RSQSim earthquake simulator. More specifically, the authors analyze how fault connectivity at depth and fault sinuosity both at surface and depth drives the magnitude frequency distribution, max magnitude and probability of surface rupture. The Mt. Vettore (Central Italy) area is taken as a test area, to run the models and compare the outputs with coseismic and long-term slip.

I enjoyed reading the paper and I want to congratulate the authors for putting together such interesting research. The text is properly organized, figures are illustrative and well-detailed in the text. The paper is of high significance, since it applies earthquake cycle simulators to the evaluation of surface rupture probability; to my knowledge, it is the first attempt in the literature, and the paper could pave the road for a wider application in PFDHA, by providing an alternative approach with respect to those more commonly applied. Given the above, I suggest accepting the paper following minor reviews.

Below I list some comments which I hope may be useful for the revision stage.

Author's response (AR): We kindly thank the reviewer for the positive and optimistic comments on our manuscript and its significance to PFDHA.

Specific comments:

Lines 10-12: this sentence may benefit from rephrasing. The likelihood of surface rupture is just one of the several components needed to run a PFDHA. Overall, PFDHA estimates the likelihood of exceeding a given displacement value, usually expressed as annual frequency of exceedance.

AR: We agree. We have better detailed this in the revised version of the manuscript (lines 11-12).

Introduction: consider to add the references to the recently published IAEA Tecdoc 2092 https://doi.org/10.61092/iaea.74us-dn4n and the paper by Valentini et al 2025 (10.1029/2024RG000875)

AR: We thank the reviewer for highlighting these recent references. In the revised manuscript we have included these citations in the Introduction section (line 27).

Lines 62-63: compared to many (most of?) other active faults worldwide, the Mt Vettore fault geometry is quite well-constrained. I would not say that data on the subsurface geometry is not available, given also that the fault was responsible for the widely studied 2016 earthquake sequence.

AR: While several models of Mt. Vettore's subsurface geometry have been proposed, especially since the 2016 earthquake sequence, we recognized little consensus on a preferred geometric model. This lack of agreement is evident in studies such as Tung and Masterlak (2018), who tested multiple source geometries to explain the August 2016 earthquake, underscoring the coexistence of different models. For instance, Lavecchia et al. (2016) propose a model, (used as a reference for our segment-connected configurations), where fault segments connect at ~7km with two distinct dip domains: 60º between 0 an 8 km, and 45º below 8km. In contrast, models by Cheloni et al. (2019) or Falucci et al. (2018) assume gradual listric geometries for the Mt. Vettore fault but do not discuss how or whether fault segmentation is linked at a specific depth. This lack of consensus combined with the exceptional body of surface geological data in the region is what makes the Mt. Vettore a good candidate for our study.

In the revised manuscript, we have clarified this rationale in the Introduction section (lines 73-75).

Section 2.2. Can you provide a numerical measure of the different degrees of sinuosity in your models? Something like the sinuosity index of a river. It may help the reader to grasp the variability among models.

AR: This comment links with reviewer's #2 comment on model nomenclature. In the revised manuscript, we have adopted a new nomenclature that includes a numerical attribute to describe sinuosity: 0, 0.3, 0.6 and 1 to describe all models from minimum to maximum sinuosity, respectively. With this, we do not think it will be necessary to compute the actual sinuosity of the models because this new nomenclature is intuitive and will already help readers understand the variability. This new nomenclature is applied to all figures in the manuscript and supplements, as well as in the explicit references of models throughout the text (see manuscript with tracked changes).

Section 2.2. Why have you selected the "fault" level of the Central Apennine database instead of the "trace" level? Is it a matter of model resolution (i.e., the 300 m wide fault elements)?

AR: Yes, but not only. Model resolution is an important reason for choosing the fault level over the trace one. The trace level implies smaller order geometric complexities that would require finer meshes and a substantial increase in computational cost. More importantly, at Mt. Vettore the trace level is strongly biased toward the 2016 surface rupture locations. In those areas, the mapping is very detailed, including secondary ruptures whose depth geometries and connections remain unresolved. Incorporating such features into RSQSim would require speculative assumptions about their subsurface relationships, introducing subjectivity into the model. With the fault level we achieve a better balance between model activity related to the main fault structure (not secondary, smaller ruptures), while preserving the geological segmentation described in literature.

Section 2.3.3. Several slip rate estimates at surface are available for the fault segments from Cupi to Mt Vettore; they cover different time intervals (e.g., post-galcial; long-term geological). Since you need to define the slip distribution to run the model, I'm wondering if considering a distribution directly derived from the surface measurements along the entire 40-km long system (instead of the distribution detailed at lines 163-170) could make an impact on the obtained results.

AR: The reviewer raises a very good point in this comment. Prescribing a more complex slip-rate distribution could indeed influence the results, though we expect the effect to be limited. Slip rate primarily governs: 1) the locations where earthquakes preferentially nucleate over long time spans (that is why we our use a tapered distribution at the fault edges to promote nucleation at realistic seismogenic depths), and 2) the total slip on each patch, which should converge to the prescribed value by the end of the seismic cycle (assuming full coupling).

In our case, modifying slip rate at surface would mainly affect the long-term cumulative slip distribution along-strike, which would reproduce the one prescribed. However, slip rate is not the sole control on cumulative slip. As shown in Fig. 12, despite imposing a parabolic slip-rate profile along-strike, the long-term cumulative throw distribution for Mw ≥ 5.5 events is strongly influenced by geometric features such as segmentation and bends, which locally reduce throw values. This indicates that a simplified input distribution already reproduces most of the observed cumulative throw patterns.

Introducing more complex slip rate distributions from surface measurements might slightly improve the fit between model and observations shown in Fig. 12, but it would also make it harder to isolate the effect of geometry on the resulting throw distribution. This would compromise independence between model and observations for comparison.

Line 225: I had some difficulty in understanding the meaning of Mmax in your condition rule. One needs to know that the Mmax obtained in the models is always lower than 6.6 (i.e., M2016 + 0.1), however this info is provided later in the text

AR: We agree with the reviewer and we have further clarified this in the text of the revised manuscript (lines 240-244) where the condition rule is specified. In detail, we have introduced that we use Mmax as a constrain because the maximum magnitude of the models is always around or below the Mw of the 2016 mainshock, and we have referred to the specific section of the text (section 3.1.3) where we explain the Mmax of each catalogue.

Line 229: delete one "them"

AR: We have fixed this in the revised manuscript.

Line 375 (figure 10): nice image, thanks for providing this figure. I found myself moving back and forth between figures 9 and 10, to compare the regressions and the along-strike variation in rupture probability. I found it quite interesting such comparison. For instance, in the connected constant models, the trace-linear configuration seems to have a higher probability in fig 10; however, in fig 9, at Mw 6.0 the trace-trace model lies above the trace-linear. Consider the possibility to add in figure 10 a label or colored dot representing, for each model, the probability of surface rupture at Mw 6.0 extracted from figure 9.

AR: We thank the reviewer for the positive feedback on the figure. In our opinion, adding the probability of Mw 6.0 can be confusing, because figure 10 aggregates probability for magnitudes >=6.0, while in figure 9 probabilities are bin-specific. The discrepancy between probabilities in both figures is an effect of the regression fit to the data points. In figure S4 of the Supplements we show the actual data points used to fit the regressions. In the connected constant trace-linear model, the data points of Mw 5-6 show a drop in the probabilities which overall lower the regression in comparison to the trace-trace model (see figure S4). However, if we look at these data points, we actually observe that the trace-linear model has

higher probability for Mw 6.0 than the trace-trace model. This is consistent with figure 10, which shows the actual data points, not a regression fit.

Lines 380-385: the surface rupture probability depends on the fault area; in the disconnected model, this is proportional to fault length. In figure 10, Mt Bove shows the highest probability. Does this depend on the adopted Mw-area relation? I mean, using the Thingbaijam et al relation, a Mw 6.0 corresponds to roughly 200 km2; with the 12-km seismogenic thickness, it means a ca. 15-km long fault. Is this the size of Mt Bove segment? Is the Mt Porche area (disconnected linear-linear model) enough to generate a Mw 6.0 event?

AR: Earthquake simulators do not use empirical scaling laws to derive magnitudes from fault parameters. Instead, they rely on physical equations that integrate rock friction and slip rate to model tectonic loading, nucleation and rupture throughout the earthquake cycle. In the simulations, the magnitude-rupture area relation emerges from the input fault frictional properties (e.g., rate and state parameters, initial stress conditions). In our study, we verify that such simulated Mw-area relation is consistent with empirical scaling laws, but the relation is not part of the computation.

Having said that, the Mt. Bove shows the highest probability because it is in the central section of the fault where slip rate is greatest, leading to more frequent nucleations (see figure 3a). The seamless probability transition between the Mt. Bove and Cupi Ussita segments indicates that Mw>=6 often propagate across these two segments. Conversely, the sharp probability drop at the Mt. Porche segment suggests that Mw>=6 do not propagate into or out of this segment.

The following figures show the GR distributions for each segment in the Disconnected Linear–Linear and Trace–Trace models, along with the frequency of single- and multi-segment ruptures for Mw ≥ 6. Single and multi-fault ruptures at the Mt. Porche are consistently the less frequent. This likely reflects 1) its geometric configuration, preventing rupture propagation, and 2) its smaller size compared to the other segments. Regarding the reviewer's question on segment size: in the Linear–Linear model, Mt. Porche does not generate any Mw ≥ 6 ruptures.

[Figure]

Left column: Disconnected Linear-Linear (D0) / Right column: Disconnected Trace-Trace (D1)

Line 469: here you highlight the importance of constraining fault traces at surface. In some cases, different interpretations may be present. For instance, the CAD fault database is the result of a big effort in data harmonization by several groups which may have mapped the fault traces in a slightly different manner. Do you have any hint on how to incorporate such uncertainty into the modeling setup?

AR: Earthquake simulators are deterministic in terms of model set up and constraints, meaning that uncertainties are not intrinsically explored. Similar to what we do with subsurface geometry, one could indirectly explore such uncertainties by proposing a suite of different fault trace hypotheses and perform a sensitivity analysis to determine their impact into the results. For instance, an analysis like the one we mention is performed by Zielke and Mai (2025). In order to incorporate these epistemic uncertainties, a logic tree approach could be implemented were several conflicting interpretations are available.

Line 494: typo in broadest

AR: We have fixed this in the revised manuscript.

Lines 536-540: another factor at play to explain the lower probabilities obtained by empirical models could be the role of local properties and in particular near-surface materials. Loose sediments or weak rocks favor an accommodation of slip by tilting/warping rather than brittle fracturing.

AR: This is a very interesting point. We have added it to the discussions in the revised manuscript (lines 574-577).

I acknowledge that it is beyond the scope of the paper, but as a side note I think it may be interesting to investigate the amount of surface slip in your surface rupturing events, and to see to which extent retaining only events with slip higher than (say) 5 cm moves the regressions toward higher magnitudes.

AR: This is an important point that we considered during the development of the study. We chose not to impose a slip threshold to avoid introducing arbitrariness in its selection. Moreover, damage-relevant offsets vary across infrastructures and engineering applications, and since our work has a scientific rather than engineering focus, we decided against including a threshold.

We attach the regressions of Fig. 9a computed with four slip thresholds (5 cm, 10 cm, 20 cm, 30 cm). The regressions are identical for all cases except for the 30 cm threshold, which is, obviously, only considered here for comparative purposes. The stability of the regressions, even at relatively large slip thresholds (i.e., 20cm), indicates that the simulated surface rupture behaviour is realistic, and does not produce unrealistically small surface slips.

[Figure]

Line 593: in figure 8 you consider earthquakes with Mw > 4.0; in figure 10 Mw > 6; in figure 12 Mw > 5.5. I understand the reasoning behind such choices, but explaining this aspect earlier in the manuscript could enhance clarity.

AR: Agreed. We have clearly specified the rationale between the different magnitude threshold selections in section 2.5 (lines 212-215), where we explain the methods conducted for the catalogue analysis.

Section 4.5. I agree that earthquake simulators can overcome some of the limitations of empirical datasets. However, the method applied in this paper requires quite detailed site-specific data, which may not be always available: do you think this aspect can limit the applicability of earthquake simulators for PFDHA studies? For instance, the earthquake approach in PFDHA is much more used than the displacement approach (Youngs et al 2003).

AR: We thank the reviewer for raising this point. Our study focuses on a region with abundant site-specific data, which we used primarily to evaluate how well our simulations reproduce observations and surface rupture behaviour. We agree that limited site-specific data can constrain the development and validation of any methodology, including ours. However, most of the datasets we use are not required to run the simulations but to assess their performance. Therefore, the applicability of earthquake simulators is not inherently restricted by the availability of detailed local datasets. On the contrary, in cases with scarce site-specific data (e.g., single-event displacement datasets for the displacement approach), earthquake simulators may provide an alternative to the earthquake approach in PFDHA. As shown in this study, simulator tools may also enable uncertainty and parameter sensitivity exploration, with large datasets that can strengthen statistical analyses.

That said, as in any hazard study, a minimum level of information is necessary to properly constrain the simulations and avoid speculative assumptions (e.g., fault slip rates, fault mapping and geometry). In the revised manuscript, section 4.5 (lines 699-701) we have clarified this limitation and better defined the role of earthquake simulators within this framework.

Lines 643-644: Fault-specific analyses in PFDHA are better addressed with the displacement approach rather than the earthquake approach.

AR: Noted. Simulators provide the ability to generate populated rupture datasets with high resolution fault displacement data, which would enable fault-specific analyses with the displacement approach. We have specified this better in section 4.5 (lines 688-690).

References mentioned:

Cheloni, D., Falcucci, E., Gori, S.: Half-Graben Rupture Geometry of the 30 October 2016 Mw 6.6 Mt. Vettore-Mt. Bove Earthquake, Central Italy, Journal of Geophysical Research: Solid Earth, 124, 4091–4118. https://doi.org/10.1029/2018JB015851, 2019

Falcucci, E., Gori, S., Bignami, C., Pietrantonio, G., Melini, D, Moro, M., Saroli, M., Galadini, F.: The Campotosto seismic gap in between the 2009 and 2016–2017 seismic sequences of central Italy and the role of inherited lithospheric faults in regional seismotectonic settings. Tectonics, 37, 2425–2445. https://doi.org/10.1029/2017TC004844, 2018.

Lavecchia, G., Castaldo, R., De Nardis, R., De Novellis, V., Ferrarini, F., Pepe, S., Brozzetti, F., Solaro, G., Cirillo, D., Bonano, M., Boncio, P., Casu, F., De Luca, C., Lanari, R., Manunta, M., Manzo, M., Pepe, A., Zinno, I., and Tizzani, P.: Ground deformation and source geometry of the 24 August 2016 Amatrice earthquake (Central Italy) investigated through analytical and numerical modeling of DInSAR measurements and structural-geological data, Geophysical Research Letters, 43, https://doi.org/10.1002/2016GL071723, 2016.

Tung, S., Masterlark, T.: Resolving Source Geometry of the 24 August 2016 Amatrice, Central Italy, Earthquake from InSAR Data and 3D Finite-Element Modeling. Bulletin of the Seismological Society of America, 108(2): 553-572, https://doi.org/10.1785/0120170139, 2018.

Zielke, O. and Mai, P. M.: Does Subsurface Fault Geometry Affect Aleatory Variability in Modeled Strike-Slip Fault Behavior?, Bulletin of the Seismological Society of America, 115, 399–415, https://doi.org/10.1785/0120240152, 2025.

**Responses to Referee #2**

The probability that an earthquake becomes a surface-rupturing event is a key ingredient in probabilistic displacement hazard analysis. Robust estimates of these probabilities are limited by the scarcity of surface rupturing events. Understanding how different fault properties, such as geometry, connectivity at depth, or sinuosity, affect this probability, is hampered by the lack of detailed observations at depth. Gomez-Novell et al. bring an innovative approach to this data gap. They use rupture simulators to test the effect of different fault geometries at depth on the probability that an event becomes a surface rupturing one on the Mt Vettore fault in Italy. Their study highlights how geometry influences the probability of surface rupture and offers a pathway to incorporate inferences from simulators into PFDHA. The contribution is original and useful and I support eventual publication.

Author's response (AR): We kindly thank the reviewer for the very positive feedback on the article.

I have some minor comments that are mostly focused on improving the clarity of the article:

Figure 2: the segmentation and smoothness degrees the authors test are very reasonable but I find the trace-trace trace-smooth etc. wording to be very confusing. I think the suite of geometries may be captured by two constraints: a segmentation (n of separate segments) constraint, and a roughness (for example, RMS roughness as used in fault roughness studies). These would describe the suite of geometries quantitatively and remove the confusion infused by the naming choices.

AR: This comment highlights a confusing selection on model nomenclature that we agree with and, as such, we have taken appropriate measures to solve it in the revised manuscript. In detail, we have adopted a parametrized nomenclature. Letters for the connectivity level: D- Disconnected, C-Connected constant and L-Listric; increasing numeric values for increasing sinuosity/roughness of the fault, going from 0 (minimum sinuosity) to 1 (maximum sinuosity). As such, the models have become:

- Disconnected: D0 (Linear-linear), D0.3 (trace-linear), D0.6 (trace-smooth), D1 (trace-trace).
- Connected constant: C0, C0.3, C0.6, C1
- Listric: L0, L0.3, L0.6, L1

We believe this nomenclature for the sinuosity is more intuitive, helping readers to clearly understand the models that have higher or lower sinuosity, without the need to numerically compute it from the models. In the manuscript text (section 2.2; lines 120-130), we have provided a proper introduction to this new nomenclature, and we have replaced all explicit model mentions to the former nomenclature with the new one (figures and text).

Line 229 typo in extra them

AR: We have fixed this in the revised manuscript.

Figure 7 - I don't understand why the connected listric would produce larger magnitude events than the connected constant - isn't the listric geometry quite unfavorable for slip propagating into those regions?

AR: The listric model produces larger magnitude events than the non-listric because the fault rupture area available is larger. Even though the listric geometry causes dip variations at depth, these dip variations are not very large and quite gradual. Consequently, the fault plane stresses likely do not show sharp transitions at depth enough to prevent ruptures from propagating.

Figure 8 - telling these models apart visually is a bit hard. Can the authors fit a logistic regression to highlight the differences between the two end-member models? Should be easy to do since the authors do it anyway to provide the parameters in the next table and have the regressions in Fig 9.

AR: The purpose of this figure is to show that earthquake rates for a given time period do not correlate with surface rupture probability, which is evident when looking at the magnitude bin-specific probabilities. While a logistic regression might improve the visualization, it would represent a fit to the data rather than the data itself. Such a fit can deviate from the actual values and potentially mask the rate-independence relationships we want to highlight.

Line 327 - the authors point out that a and b are not the rate and state friction coefficients but the intercept and slope of the logistic fits. This is a useful consideration. They should also point out that a and b are not the parameters in the magnitude-frequency distribution, since this is another possible source of confusion given the nature of the article.

AR: Agreed. We have added this clarification in the revised version in the manuscript (lines 350-351).

Figure 10 - consider not using a divergent color map, since the probabilities go from 0 to 1.

AR: We have considered this for the revised manuscript by adopting a perceptually uniform sequential colormap ("plasma"), as we show below.

[Figure]

I appreciate how this article weaves the modeling results with the results from empirical studies in the literature.

AR: We are thankful for the positive feedback on this comment.

The authors could refer to Valentini et al. (2025)'s call for more model-driven advances to supplement current PFDHA approaches as part of the justification for this work.

AR: We agree and we have added this in the revised manuscript version, specifically in the introduction (lines 44-46) and in section 4.5 (lines 691-692).

**Responses to Referee #3**

Dear editor,

Thanks for the opportunity to review "Coseismic Surface Rupture Probabilities from Earthquake Cycle Simulations: Influence of Fault Geometry" by Gomez-Novell et al. I enjoyed reading the paper; it is well written and makes some interesting points, and in my opinion is worth publishing.

Author's response (AR): We are thankful for the positive feedback on the paper.

I have a few high-level comments that — if addressed — could help significantly improve the paper:

This isn't the first paper to consider RSQSim as a useful tool for PFDHA... The authors should check out Daglish et al. (2025). The present study is much more local in focus and considers impacts of fault geometry, which Daglish et al. do not consider. However, some of the discussion by Daglish et al. may be useful to the authors, especially in the light of my other comments.

AR: We thank the reviewer for pointing out this article, which went unnoticed to us. We have acknowledged this interesting research in two sections of the revised manuscript:

1. Introduction: We noted that Daglish et al. (2025) is the first study to apply simulators to FDHA applications and highlighted the differentiating aspect of our approach to better situate our work within the PFDHA research field (lines 50-59).

2. Discussion: Throughout section 4.5, we have emphasized how the results by Daglish et al. (2025) reinforce the reliability of earthquake simulators for PFDHA, and also the limitations acknowledged in the paper.

The manuscript is overwhelmingly positive about the potential of earthquake simulators for PDFHA... Some detailed discussion of limitation (at least 1-2 paragraphs) would give a more balanced view. For example, I think it's important to discuss uncertainties in trace location, distributed off-fault deformation and the challenges associated with identifying a primary trace.

AR: Throughout the manuscript discussions we acknowledge several limitations of our study. In section 4.2 (lines 474-477 of the original preprint), we discuss how uncertainties in fault traces might be addressed by exploring multiple realizations of fault geometries in line with previous research by Zielke and Mai (2025). We agree that the impacts of not considering off-fault deformation, distributed rupturing or the challenges associated with identifying a primary trace are not addressed in our manuscript. To accommodate this, we have:

1. Detailed the implications of not considering off-fault deformation in section 4.3 (lines 578-583). Our study only tackles one part of PFDHA (primary surface ruptures), which can carry potential underestimations of hazard associated to not considering off-fault deformation and distributed rupturing. In the discussion we have also proposed solutions o how these limitations could be mitigated. For instance, the simulated displacements on the principal fault from RSQSim could be used along with approaches like the one from Visini et al. (2025) or Daglish et al. (2025) to develop probabilistic models of distributed fault ruptures.

2. Discussed challenges of identifying primary traces in section 4.2 (lines 508-513). In relation to what we discuss in section 4.2, we have further highlighted the importance of exploring several fault trace hypotheses to capture such uncertainties in the hazard assessments.

The authors are quite evangelical about RSQSim as a simulator and could (ideally) tone down their language slightly throughout the manuscript. I agree that RSQSim does a surprisingly good job generating realistic-looking populations of synthetic earthquakes considering how much it simplifies earthquake physics, but it is only a simple model and is definitely missing some aspects of realistic earthquake slip distributions. The authors should include discussion of the limitations of RSQSim, especially for generating earthquakes.

AR: In section 4.5 we acknowledge that earthquake cycle simulators like RSQSim have important limitations in the physical representation of earthquake rupture processes, especially when compared to fully dynamic rupture simulations (though the comparison is not quite fair, as both approaches serve different purposes). We also discuss in section 4.3 the impact that model parameters such as a-b coefficients or initial stresses have on the simulations. The main advantage of using RSQSim is the balance between computational efficiency and its ability to generate realistic synthetic catalogues, both

in terms of long-term statistics and rupture characteristics. We demonstrate this good performance by comparing the simulated catalogues with empirical relations and the simulated coseismic and cumulative slips with field observations. Naturally, discrepancies remain, but these can also reflect processes that have not been considered in the modelling (e.g., erosion, soft sediments, smaller order fault complexities)

Most limitations of RSQSim are already detailed in the original publications by its developers. In the revised manuscript we have moderate some statements on RSQSim's advantages and have expanded the discussion on model limitations in section 4.5 (lines 695-709).

I know the focus is on the influence of fault geometry, but I think the study should be expanded significantly to understand the sensitivity of surface rupture probabilities to prescribed slip distribution and rake, and the relative importance of those factors compared with fault geometry. For example, I think that the assumed — and largely unconstrained — slip-rate distribution will potentially influence the modelled earthquakes more than geometry. I couldn't find any indication of what rakes the authors specified, but I assume pure normal... Setting a constant horizontal azimuth of extension and adjusting rakes to match that azimuth could also make a big difference to modelled earthquakes. I think those factors are really worth exploring... It is a bit of work but I think it's important and not enough for a separate paper.

AR: We agree with the reviewer that exploring the sensitivity of these parameters would be very interesting, but we do not think it is pertinent for the present paper:

1. The focus of the paper is to evaluate (and isolate) the impact of fault geometry on surface rupture probabilities. Adding more variables to the study, in its current form, will imply major re-design of the modelling set-up and the subsequent analyses. On the one hand, we would necessarily have to enlarge the model sample to accommodate further parameters into the exploration tree. On the other hand, we would not be isolating the effect of geometry anymore. Instead, we would be combining effects and potentially inducing interactions between parameters that might not be trivial to analyse. For instance, some parameters can show non-linear relationships that are not easily identifiable nor quantifiable with the current experimental design. We think these relationships would be rather determined with more advanced techniques like machine learning algorithms (e.g., random forest) that are out of the current scope.

2. The slip rate distribution is not unconstrained, it is informed by surface geological data and follows a tapered pattern towards the fault edges, consistent with observations in the Apennines. Our simulations show that slip rate has a limited effect compared to geometry. For instance, in all connected models, nucleation occurs at depths of segment linkage regardless of slip-rate maxima being deeper. Moreover, the adopted distribution does not prevent long-term slip behaviour to show features that are consistent with observations (e.g., Figs. 11 and 12). While alternative slip-rate distributions could generate differences in the models, our results suggest that their influence is less significant than the reviewer anticipates. The effect of slip rate variability in the simulations is a very interesting and pertinent point, but including it in our study is not pertinent because it would imply major changes in the current experimental design and analyses, and a significant lengthening of the paper. We think that exploring the effect of slip rate can stand on its own as future work, especially considering the uncertainties of this parameter in many regions. In fact, constraining slip rate is a central topic in most fault-based seismic hazard studies, and a dedicated study on this aspect could help address these issues more clearly.

3. We recognize the reviewer's point on the azimuth-adaptive rake; however, we consider it more appropriate not to include it in our analysis, for two main reasons. The first one is that structural studies (e.g., Iezzi et al., 2018) indicate that the slip vector of the Mt. Vettore Fault during the 2016 surface ruptures is for the most part perpendicular to the fault strike along the fault trace, only oblique in the northern sector of the Vettoretto Redentore segment. Second, the exploration of a geometry-dependent rake would generate significantly different rake fields across the models. These model differences could interfere with strictly geometry-related features in the catalogue, ultimately interfering with the analyses that we present.

I'm worried by the way slip-rate distributions are specified... Delogkos et al. (2023) tapered slip towards the edges of each fault segment, whereas this study tapes across the whole fault segment. I think that combined with the loading scheme, this approach will potentially lead to nucleation of very large (unrealistic) numbers of small earthquakes close to fault edges. That effect may be negated in the analysis by the minimum of 10 patched that the authors impose, but it is important to provide better visualisation of where earthquakes nucleate in the model, either in the main paper or supp info.

AR: We appreciate the reviewer's concern with the slip rate distributions adopted in our study. There are a few key points that justify our decision:

1. Consistency across models: Applying a whole fault tapering approach ensures comparability between the different geometric configurations explored. If we tapered at the edges of individual segments, this would lead to significantly different slip rate distributions between disconnected and connected models – particularly at depth. Different slip distributions across the fault planes would necessarily produce divergences in peak values, since they are scaled so that the average for all fault elements equals 1 mm/yr. This could introduce variability between catalogues linked to the imposed slip distribution, potentially obscuring the variability that is strictly geometry-related.

2. Geological constraints: Linking with our response to reviewer #1, the slip rate distribution is designed to reflect surface geology (e.g., 1mm/yr in the central part of the fault), while keeping it simple enough to avoid overfitting to localized slip complexities recognized in literature, which often reflect site-specific effects (e.g., fault bends).

3. Focus on geometry control: One of our aims was to test whether the observed along-strike slip variability, like cumulative slip reduction at the segment tips, could emerge from fault geometry alone. A simplified slip rate distribution allows us to isolate geometric influence on the final slip of the models.

4. Consistency with higher order structure: The Mt. Vettore fault, although segmented, is regarded as a single higher order fault structure. Even though the relationship of these segments at depth is not clear, we wanted to adopt a slip rate distribution that reflects the large-scale structure of the Mt. Vettore fault, rather than imposing slip rate tapering that we do not know how resolves at depth.

As the reviewer anticipates, there are earthquake nucleation artifacts at the segment tips in our models. But these have a limited impact in our analysis.

A. Impact on surface rupture regressions:

Most of these nucleations correspond to low magnitude earthquakes (Mw 4-5), as illustrated in the figure A (see below) for the two end member connectivity models. Their frequency decreases markedly for Mw>=5, and since Mw<5 earthquakes rarely generate surface ruptures, the effect of such artifacts on the regressions is likely unnoticeable.

In addition, hypocentral depths of these artificial Mw 4-5 nucleations (6-9 km; Fig. B) are consistent with depth distributions of real earthquakes in the Apennines. This indicates that shallow, fault-edge nucleations are proportionally uncommon, further limiting any potential bias in the surface rupture probability regressions (depth of nucleation is a big factor controlling surface rupture likelihood).

B. Impact on spatially variable surface rupture probabilities:

Spatially variable surface rupture probabilities for Mw $\geq$ 6 (Fig. 10 of the paper) are primarily controlled by the along-strike slip rate distribution and fault geometry. Local probability increases at segment boundaries may partly reflect minor nucleations near fault edges (Fig. A, right column), but they are also explained by geometric complexities (such as bends or deep segment connections) that promote earthquake nucleation (also seen in Fig. A). This is consistent with previous observations; for instance, Lavecchia et al. (2016) documented that the 2016 Mw 6.0 Amatrice earthquake nucleated at an intersegment zone where two faults link at depth.

Importantly, the key outcome of PFDHA is the probability of exceeding a slip value at a site over a given time span, which depends on that site's slip history. Thus, even if surface rupture probabilities are elevated at the segment edges, the hazard is ultimately governed by the coseismic slip recorded in these regions (especially with the displacement approach). Our models show that both coseismic and cumulative slip taper toward segment edges (Figs. 11 and 12), independent of nucleation location. This suggests that localized increases in surface rupture probability are unlikely to bias fault displacement hazard, since these regions contribute less slip.

In the revised version we have included the figures shown here as Supplementary information (figures S4 and S7), with the corresponding renumbering of supplementary figures. In the revised manuscript we have also 1) added the justification on the slip distribution in section 2.3.3 (lines 170-177) and 2) added the discussion described above on anomalous earthquake nucleation in section 4.2 (lines 469-483).

[Figure]

Fig. A

[Figure]

Fig. B

References mentioned:

Daglish, J. M., Stahl, T., Howell, A., and Wotherspoon, L.: Advancing regional analysis of road infrastructure exposure to fault displacement hazard: A New Zealand case study, International Journal of Disaster Risk Reduction, 122, 105440, https://doi.org/10.1016/j.ijdrr.2025.105440, 2025.

Iezzi, F., Mildon, Z., Faure Walker, J., Roberts, G., Goodall, H., Wilkinson, M., Robertson, J. Coseismic throw variation across along-strike bends on active normal faults: Implications for displacement versus length scaling of earthquake ruptures. Journal of Geophysical Research: Solid Earth, 123, 9817–9841. https://doi.org/10.1029/2018JB016732, 2018.

Lavecchia, G., Castaldo, R., De Nardis, R., De Novellis, V., Ferrarini, F., Pepe, S., Brozzetti, F., Solaro, G., Cirillo, D., Bonano, M., Boncio, P., Casu, F., De Luca, C., Lanari, R., Manunta, M., Manzo, M., Pepe, A., Zinno, I., and Tizzani, P.: Ground deformation and source geometry of the 24 August 2016 Amatrice earthquake (Central Italy) investigated through analytical and numerical modeling of DInSAR measurements and structural-geological data, Geophysical Research Letters, 43, https://doi.org/10.1002/2016GL071723, 2016.

Visini, F., Boncio, P., Valentini, A., Scotti, O., Nurminen, F., Baize, S., and Pace, B.: Empirical regressions for distributed faulting of dip-slip earthquakes, Earthquake Spectra, 87552930241308860, https://doi.org/10.1177/87552930241308860, 2025.